## Registered report

 

Subject Areas:
neuroscience/psychology

Keywords:
natural scene perception, contents of consciousness, massive report paradigms, consciousness, qualia, expectation

Author for correspondence:
Liang Qianchen
e-mail: qianchen.liang@monash.edu

# How much can we differentiate at a brief glance: revealing the truer limit in conscious contents through the massive report paradigm (MRP)

Liang Qianchen[1,2], Regan M. Gallagher[1,2] and Naotsugu Tsuchiya[1,2,3,4]

[1]School of Psychological Sciences, Faculty of Medicine, Nursing and Health Sciences, Monash University, Clayton, Victoria, Australia
[2]Turner Institute for Brain and Mental Health, Monash University, Melbourne, Victoria, Australia
[3]Center for Information and Neural Networks (CiNet), Osaka, Japan
[4]Advanced Telecommunications Research Computational Neuroscience Laboratories, Kyoto, Japan

LQ, 0000-0002-5415-9223

Upon a brief glance, how well can we differentiate what we see from what we do not? Previous studies answered this question as 'poorly'. This is in stark contrast with our everyday experience. Here, we consider the possibility that previous restriction in stimulus variability and response alternatives reduced what participants could express from what they consciously experienced. We introduce a novel massive report paradigm that probes the ability to differentiate what we see from what we do not. In each trial, participants viewed a natural scene image and judged whether a small image patch was a part of the original image. To examine the limit of discriminability, we also included subtler changes in the image as modification of objects. Neither the images nor patches were repeated per participant. Our results showed that participants were highly accurate (accuracy greater than 80%) in differentiating patches from the viewed images from patches that are not present. Additionally, the differentiation between original and modified objects was influenced by object sizes and/or the congruence between objects and the scene gists. Our massive report paradigm opens a door to quantitatively measure the limit of immense informativeness of a moment of consciousness.

# 1. Introduction

How informative is a moment of conscious experience? While this question sounds simple, different interpretations of the terms in this sentence have caused much confusion and many debates and controversies among philosophers and scientists. Throughout the paper, by 'consciousness' we mean the 'contents' of our subjective experience (i.e. 'what it is like'), and not the overall level of consciousness [1,2].

From a viewpoint that emphasizes what participants can report, plan for behaviours or use for reasoning, only a small number of percepts would qualify as consciously accessible at any given time [3]. This view has been informed and supported by empirical evidence, such as change blindness [4,5] and inattentional blindness [6,7]. In these cases, people can miss highly salient information in the environment if they are not in the focus of attention. As a result, attention is often described as a cognitive mechanism that selects the information for conscious access [8–11].

Distinct from such view, others claim that perception is in fact highly informative but we can report only a fraction of what we consciously experience at any given time [12]. There are many possible mechanisms that can explain why we cannot report all of what we consciously experience at any given time. For example, attention may not be necessary for consciousness, but it may be necessary for subsequent conscious access or recall for reports.

In this study, we take a neutral view on this debate *per se*. However, we question the experimental design features that are commonly employed in the studies that tend to support the former viewpoint. Specifically, we question if the restrictions in the variability of the stimuli and reports that participants can make might have led to a substantial underestimation of what we can consciously experience and report.

## 1.1. Massive report paradigm

As a potential remedy to the limit in reportability and stimulus variability, we propose a novel paradigm, which we term the massive report paradigm (MRP). Using MRP, we will characterize the informativeness of consciousness, by quantifying an experiencer's accuracy in discriminating 'what they saw' from 'what they did not', expanding the number of alternatives massively [13]. If consciousness is rich, then the experiencer should be able to differentiate one experience from a huge number of other possible experiences.

Consider a case where you report seeing red, upon seeing a red patch. It seems obvious that such a simple report does not exhaust your experience at that moment. If you were allowed to report anything that you experienced, then you might have reported a particular shading of red, the spatial extendedness of the patch, shape of the patch, location of the patch, complex tinge on the surrounding, the apparent texture of the patch and its relation to any surrounding environment and so on [14]. In most experimental paradigms, however, the alternatives that you would be given for report options are severely constrained (e.g. choosing one colour out of many colour options). Yet, it seems obvious that you 'can' report more about the experience if there are more options. For example, upon seeing a red colour patch, you may not agree with seeing any of the millions of picture frames from all movies.

By expanding both reports and stimuli, the MRP will make a first step towards estimating the truer extent of informativeness in phenomenal consciousness.

More specifically, to disrupt the limitations in the reports and stimulus variability, we will use natural scene stimuli as targets and non-targets and introduce multiple responses per trial (i.e. 20 responses in Experiment 1 and 6 responses in Experiment 2) and many more participants (i.e. 240 online participants in registered Experiment 2).

## 1.2. Gist-based object recognition and informativeness of experience

Upon using the natural scene stimuli, we will address the role of the 'scene gist' in informativeness of experience. Like consciousness and information, 'scene gist' is yet another term that has no agreed upon definition. One possible definition of scene gist is 'whatever one sees and can report upon seeing it for a brief moment' [15]. Here, we adopt a more specific definition of scene gist as 'the overall semantic meaning of the scene' [16]. Taking this definition of scene gist, one extreme viewpoint would claim that all we can experience and report in a brief glance of the scene is the overall semantic meaning of the scene, but nothing more specific than that. Such a claim appears

consistent with the empirically demonstrated capacity limits, such as those found in inattentional or change blindness [4–7].

In this paper, we use a set of natural images, which are modified in terms of an object in the scene [17]. In half of the trials, a critical object is congruent with the global scene semantic (or gist) while in the rest of the trials, it is replaced with another object that is incongruent with the gist. According to the previous literature [18–23], participants were more accurate and/or faster in recognizing an object from a natural scene, when the object was congruent with the scene gist (in the sense of overall semantic meaning of the scene), compared to when the object was gist-incongruent. Therefore, we expect that the informativeness in conscious experience, that is, differentiating what is seen and not, should decrease when the object is inconsistent with the scene gist.

Upon seeing these modified natural images briefly, if participants can only experience and report the overall semantic of the scene but nothing more specific, then we should observe several notable patterns in the results. For example, we predict that upon seeing an image with a gist-incongruent object, participants should reject the gist-incongruent object in the initial image, but endorse a gist-congruent object that was absent in the image. However, if participants can consciously experience the scene beyond simple gist perception, then we predict they can accurately and consciously distinguish what was present from absent on the basis of only a short glance.

In summary, we will use the MRP to address two issues. Firstly, we will test the capacity limit of visual experience, as revealed by participants' ability to endorse present image patches, and reject absent image patches, from a briefly presented natural image. When we reduce the restriction in reports and stimulus variability, how much differentiation will participants be able to express in our task? Secondly, we will test whether or not informativeness of experience is influenced by image gists and object congruence. Can participants differentiate what they saw from what they did not when an object in the scene is congruent, but not when it is incongruent, with the overall semantic meaning of the scene? We performed Experiment 1 in the laboratory setting for $N = 15$ participants but asked 20 questions per image to achieve the massive report. In Experiment 2 (registered report), we replicated Experiment 1 in the online setting, where we reduced the number of questions to ask per participant, but we increased the number of participants ($N = 240$). With this, we make a first step towards exhausting a huge possibility of absent image patches and to collect enough data per image to estimate the truer limit in conscious visual experience in a moment.

# 2. Experiment 1

## 2.1. Methods

### 2.1.1. Participants

We obtained ethics approval from Monash University Human Research Ethics Committee before commencing this study. Fifteen participants with normal or corrected-to-normal vision were recruited via Sona Systems, an online research participant recruitment system of Monash University. Due to experimental error, we did not record the age and gender of the participants. All participants provided informed written consent prior to the beginning of the experiment.

### 2.1.2. Apparatus

We prepared the stimuli and programmed the task using MATLAB 2019a with PsychToolbox 3.0 [24]. We presented the stimuli on a DELL UltraSharp U2713Hb monitor with 2560 × 2440 pixel resolution and at 60 Hz refresh rate, which was controlled by an Ubuntu v. 14.04 Linux computer. Participants sat approximately 60 cm from the monitor to perform the task.

### 2.1.3. Stimuli

In Experiment 1, we used 82 pairs of congruent and incongruent images developed by Shir *et al.* [17]. Each image pair included an image depicting a person performing an action on an object (e.g. a girl wearing headphones; initial-congruent image in figure 1*b*), and another image in which the actioned object is replaced with an object that is incongruent with the action (e.g. a girl wearing doughnuts; initial-incongruent image in figure 1*b*). All images were cropped into a square shape, divided into 9 equal-sized patches and tagged with location numbers (figure 1*a*) to be used as test probes. Among

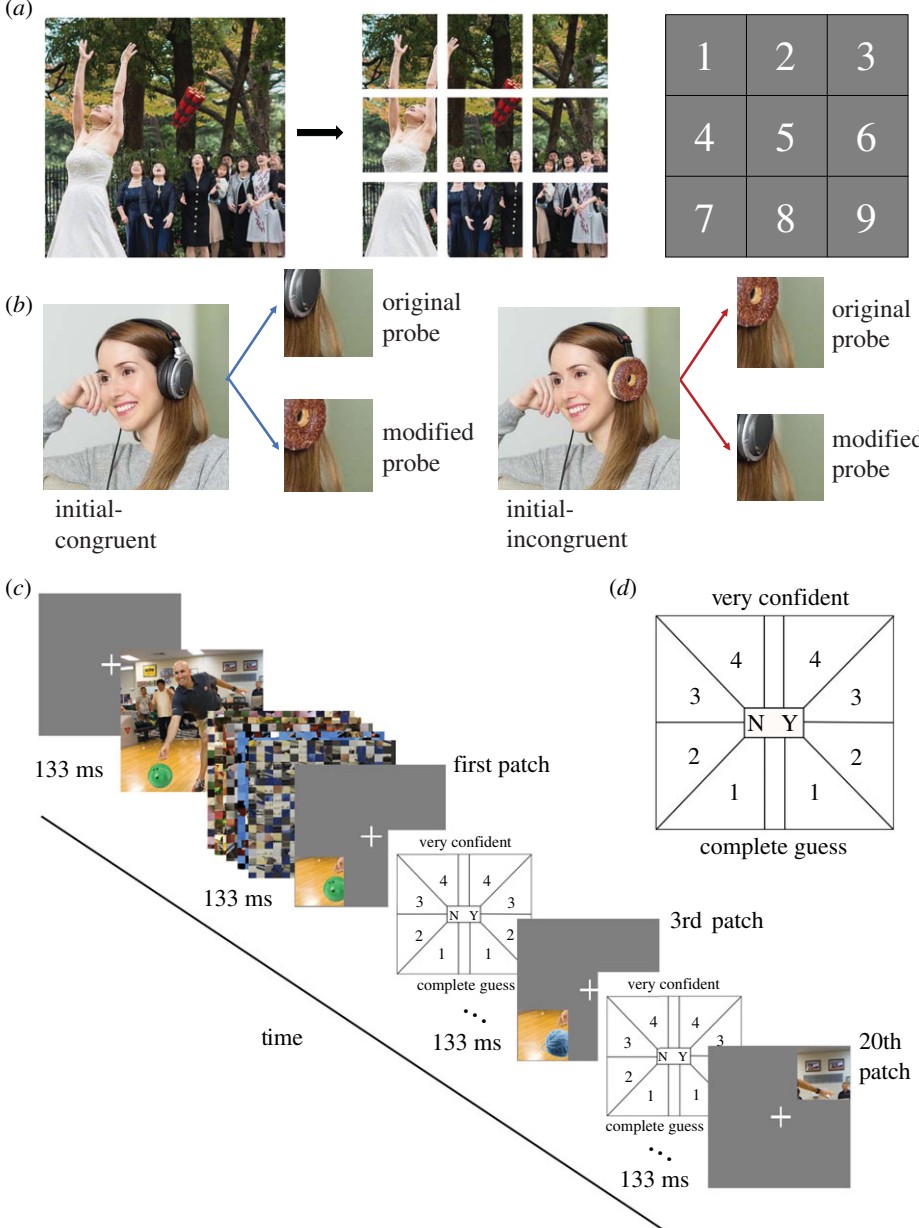

**Figure 1.** Design of stimuli and structure of a trial. (a) The procedure to generate probe patches. Each of the 9 patches were tagged with a location number 1–9 corresponding to their locations in the original image. (b) Summary of the congruence manipulation of the initial image and subsequent probe patches. The initial image is 'congruent' if it contains an object (i.e. headphone) which is congruent with the scene-gist (i.e. a girl listening to music). The initial image is 'incongruent' if it contains an object (i.e. doughnut) which is incongruent with the scene-gist. The probe object patch can be either 'modified' or 'original' with respect to the initial image (regardless of congruence of the initial image). (c) Example single-trial procedure of the experiment. At the beginning of a trial, a fixation cross appeared at the centre of the display for 500 ms. Then, either a congruent or incongruent initial image was presented for 133 ms. The image was masked by a sequence of 5 scrambled images, with each mask presented for 60 ms. After the mask images disappeared from the display, participants viewed and responded to each of the 20 probe patches (the 6 probe patches in Experiment 2). Each patch was presented for 133 ms and followed by 5 scrambled image masks of the same size as the patch. Participants had unlimited time to make a response on the response screen. (d) The response screen consists of 8 alternatives with 4 levels of confidence for both Yes and No options.

the 9 patches, we categorized the patch that contained the largest part of the object (measured in the number of pixels) as an *original object patch*, and the rest of the 8 patches as *present patches* (figure 1b). We also prepared 7044 *null patches* from a different image set, which consists of 587 natural images [25]. Sizes of all patches were the same. According to the size of the display (68.6 cm) and the

distance between the display and participants (60 cm), the size of the initial images and image probe patches was 19.4 × 19.4 degrees of visual angle (dva) and 6.5 × 6.5 dva, respectively.

### 2.1.4. Procedure

At the beginning of each trial, participants fixated on the cross presented at the centre of the screen (figure 1c). Then, an initial image (either congruent or incongruent) was presented for 133 ms, followed by 5 successive scrambled masks. After the mask, the fixation cross reappeared for 500 ms, and a test patch was presented for 133 ms, followed by another mask. If the probe was a present object patch or present patch, it was presented in the same location as it was in the initial image. When the probe was a *null* patch, its location was randomized among nine locations.

After each probe patch, a response screen appeared on the screen (figure 1d). The eight response options represented their perception regarding presence/absence of the probe patch with respect to the initial image with graded levels of confidence on their decision on a scale of 1 (complete guess) to 4 (very confident). While we did not give descriptions on the exact level of confidence that 2 and 3 represent, we told participants that larger numbers represent stronger confidence, and encouraged them to choose the number that they think describes their confidence most precisely. We did not ask participants to prioritize either the accuracy or speed giving unlimited time to make a response. Once they clicked the location of the response screen, which matched their perception best, the next probe patch was presented starting with the 500 ms fixation cross, followed by the probe, then the response screen. 20–21 probes were repeated before they went on to the next image (figure 1c).

The probe patches consisted of four types: *original*, *modified*, *present* and *null patches*. *Original* patches refer to the patches that included the critical object in the initial image. *Modified* patches refer to those that included the modified object. If the initial image was congruent, the modified patch was taken from the corresponding patch in the incongruent image (figure 1b) (vice versa for the initial incongruent image). *Present* patches were any of those that were in the initial image and not the critical patch. *Null* patches came from unrelated images (see above). The order of these four types of probe patches was randomized for each trial and participant. To avoid presenting contiguous patches across probes, we presented either five patches from location 1, 3, 5, 7 and 9 (figure 1a) or four patches from location 2, 4, 6 and 8, so that these patches include the *original* patch. In sum, a trial started with one initial image (133 ms, masked), followed by 21 (or 20) probe patches, which consisted of 1 original, 1 modified, 4 (or 3) present and 15 *null* patches. Within the trial, the order of these patch types was completely randomized for each trial and participant. To avoid any repeats of the *null* patch across 15 participants, we used 225 distinct *null* patches for a given initial image.

One trial took approximately 45 s to complete. To complete 80 trials with 40 congruent and 40 incongruent initial images, each participant attended two one-hour sessions. At the beginning of each session, they completed two practice trials, which were not included in the main 80 trials, first with the congruent, second with incongruent initial image in this order. The order of the 80 images was fixed for all participants. The congruence of the initial image was randomized across participants.

### 2.1.5. Data analysis

#### 2.1.5.1. Decision × confidence value: D × C

To represent a response for one probe patch, we encoded 'yes' decision as '+' and 'no' as '−' and multiplied it with the confidence level (1, 2, 3 and 4), to obtain decision×confidence (D × C) value, which can take one of 8 integers (−4, −3, −2, −1, 1, 2, 3, 4).

#### 2.1.5.2. Type 1 and Type 2 signal detection analysis for objective and subjective task performance

To assess participants' task performance, we employed signal detection theory [26,27] to construct a receiver operating characteristic (ROC) curve and used the area under the ROC curve (AUC) as the measure of task performance. As a measure of objective and subjective task performance, we employed Type 1 and Type 2 AUC, respectively.

#### 2.1.5.3. Objective performance: Type 1 AUC

Objective performance quantifies the discriminability of the stimuli (e.g. what was presented versus what was absent) based on confidence-weighted responses (e.g. D × C value).

To construct the ROC curve from D × C values for each participant, we defined hit and false alarm (FA) on each of the 7 decision criteria, from the strictest to the most lenient (for previous applications of this method, see [28,29]). Specifically, for the strictest criterion, we regarded a D × C = 4 to a *present or original* patch as a hit, and other responses for a *present* patch as a miss; for *null* patches, we regarded a D × C = 4 as a FA and other responses as a correct rejection. Then, we moved on to the second criterion by considering D × C = 4 and 3 as hits for *present* probes and as FA for *null* probes. We repeated this procedure until we obtained 7 pairs of hit and FA rates to construct an ROC curve.

From the ROC curve, we calculated the AUC as a non-parametric estimate of task performance. AUC ranges from 0 to 1, with AUC = 0.5 indicating chance-level performance, and AUC = 1 indicating perfect performance.

### 2.1.5.4. Metacognitive accuracy: Type 2 AUC

We also used Type 2 AUC as a non-parametric index for participants' metacognitive accuracy on the discrimination tasks. For this analysis, we treated correct trials as 'signal-present' and incorrect trials as 'signal-absent' trials. More specifically, we grouped hit and correct rejection judgement for the Type 1 analysis (i.e. 'present' judgement to a present patch and an 'absent' judgement to a null patch) as correct trials and other judgements as incorrect trials. Then, we regarded a correct trial with confidence level of 4 as a metacognitive hit, while we regarded an incorrect trial with confidence level of 4 as a metacognitive FA. We repeated this procedure until correct responses with confidence of 1 to 3 were categorized as hits, and incorrect responses with confidence of 1 to 3 were categorized as FAs. Finally, we obtained a Type 2 ROC curve by plotting the hit rates against the FA rates on each of the 3 criteria and calculated the Type 2 AUC by computing the area under the ROC.

### 2.1.6. Statistical analysis

#### 2.1.6.1. Linear mixed effects modelling

To investigate the effect of patch eccentricity on Type 1 and Type 2 AUCs, we used linear mixed effects (LME) modelling. Given our display specification (see above), the eccentricities for central patch (location 5, figure 1*a*), even-numbered patches, and odd-numbered patches (except for location 5) were 0, 6.5 and 9.2 dva, respectively. To investigate the effect of eccentricity (denoted as Ecc) on participants' performance (AUC), we built the following LME model on Type 1 and Type 2 AUCs, respectively:

$$\text{AUC} \sim \text{Ecc} + (\text{Ecc} \,|\, n) + (1 \,|\, n). \tag{2.1}$$

Here, we included participants ($n$) a random slope and intercept [30]. In order to investigate the role of eccentricity (Ecc) and congruence (Cong), we estimated the following LME model:

$$\text{AUC} \sim \text{Ecc} + \text{Cong} + \text{Ecc} \times \text{Cong} + (\text{Ecc} \,|\, n) + (\text{Cong} \,|\, n) + (1 \,|\, n). \tag{2.2}$$

We performed significance testing with likelihood ratio tests, where we computed the chi-squared statistics by comparing the likelihood of the full model against a model without the fixed effect [31,32].

## 2.2. Results

### 2.2.1. Response distributions

In Experiment 1, we asked $N = 15$ participants to view a natural image (133 ms), and then discriminate if subsequently presented small probe patches were either present or absent in the initial image one at a time over 20 or 21 probes. Figure 2 shows the proportion of responses (*y*-axis), as a function of decision × confidence (D × C; see Methods), sorted by the types of probes (across rows) and the eccentricity of the patch centre (across columns).

### 2.2.2. Present + original versus null patch differentiation

#### 2.2.2.1. Separate response patterns of present + original and null patches (first row in figure 2-I)

The first row in figure 2(i) suggests that our participants were able to discriminate between what was presented in the initial image (i.e. *present* and *original* probe patches) and what was not presented (i.e. null patches). Subdividing trials depending on the probe eccentricities (figure 2(i)*b–d*), we found that

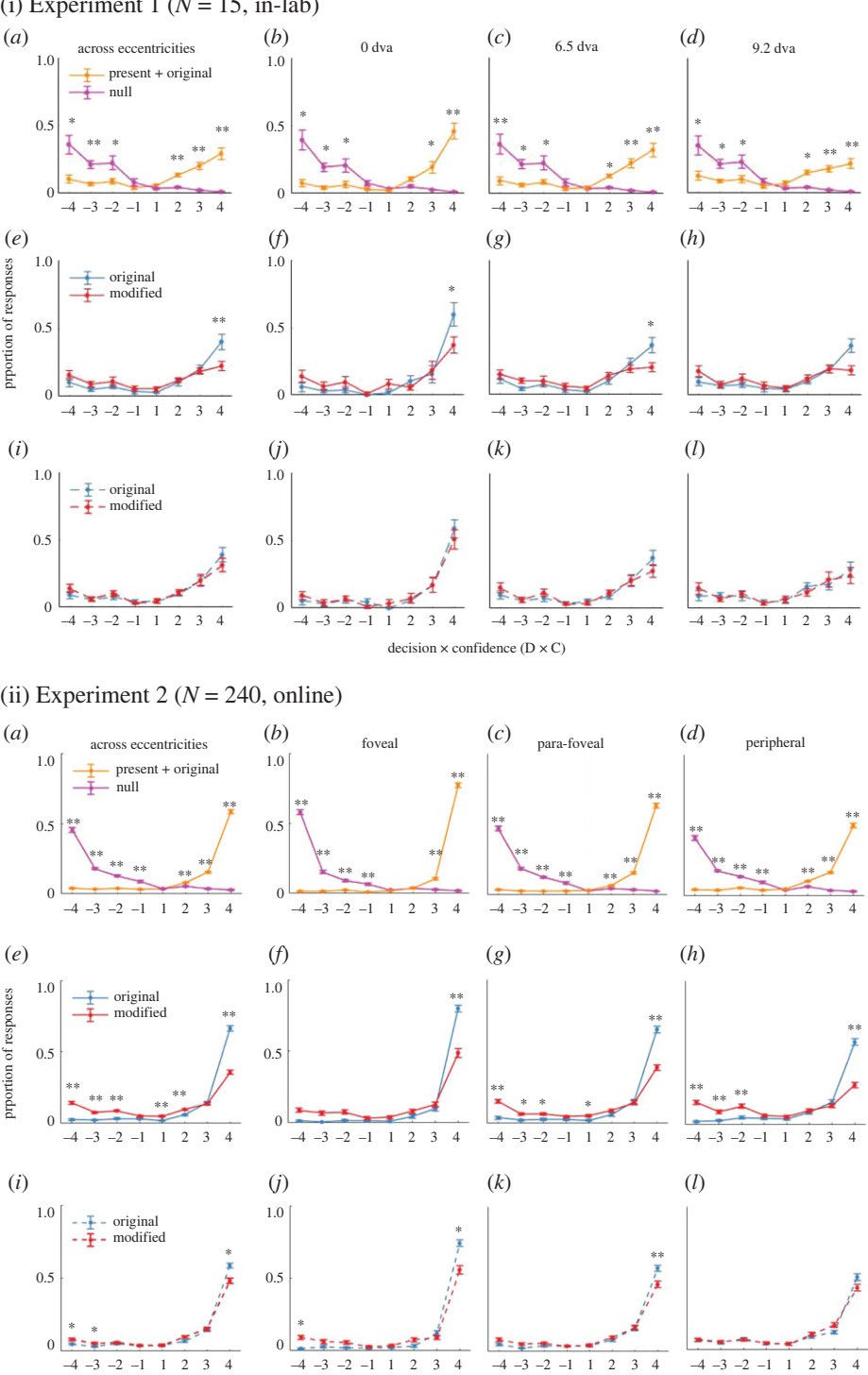

**Figure 2.** Mean percentages of responses (*y*-axis) as a function of decision ('present' = 1, 'absent' = −1) confidence (1–4) (*x*-axis) across participants in (i) Experiment 1 (top panels, in-laboratory, *N* = 15) and (ii) Experiment 2 (bottom panels, online, registered, *N* = 240). For each experiment, in each panel (*a–l*), values denoted by each colour line sum up to 1. Error bars represent standard error of the mean across subjects. Response proportion for (*a–d*) present + original patches (solid orange) and null patches (solid pink), (*e–h*) original (solid blue) and modified object probes (solid red) after the congruent initial images and (*i–l*) original (dotted blue) and modified (dotted red) object probes after the incongruent initial images. (*a,e,i*) Data pooled across eccentricities. The three right-hand columns ((*b,f,j*), (*c,g,k*) and (*d,h,l*)) represent the results with patches presented at eccentricity of 0, 6.5 and 9.2 dva for Experiment 1, and foveal, para-foveal, and peripheral for Experiment 2, respectively. We show * = *p* < 0.003 and ** = *p* < 0.0001 from *t*-tests comparing percentages of responses between present + original and null patches (*a–d*), and between original and modified patches (*e–l*), for a given D × C.

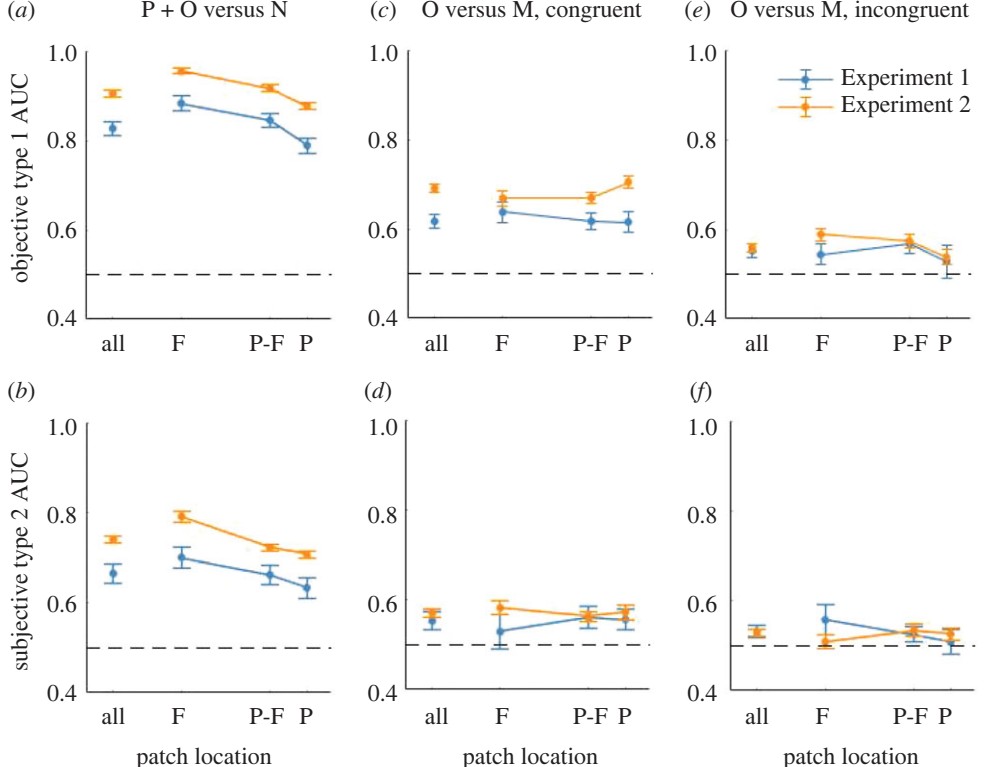

**Figure 3.** Criterion-free analyses. Objective (top row) and subjective (bottom row) task performance in two types of image patch discrimination; present + original versus null patches (a and b), as well as original versus modified patches (c and d for congruent, and e and f for incongruent initial images) in Experiment 1 (blue, $N = 15$) and Experiment 2 (orange, registered, $N = 240$). Error bars represent standard error of the mean. Dotted black lines represent chance-level (0.5) performance.

the farther the probes were located, what was present (orange lines) tended to be endorsed less while what was absent (pink lines) remained accurately rejected. The observed pattern was also supported by LME analyses: there was a significant main effect of eccentricity on the percentages of 'yes' responses on *original and present* patches ($x_1^2 = 21.37$, $p < 0.001$) but not on the 'no' responses on *null* patches ($x_1^2 = 0.01$, $p = 0.94$).

### 2.2.2.2. Objective performance (Type 1 AUC) on present + original versus null discriminability

We used criterion-free signal detection measures of objective task performance to quantify the above observations. For present versus null patch discrimination, across eccentricities, Type 1 AUC was 0.82 (±0.15, standard error of the mean; figure 3a: blue line). As shown by figure 3a, Type 1 AUC declined as a function of eccentricity, which is supported by the significant main effect of eccentricity in LME modelling results (table 1). This is consistent with the declining positive endorsement for present probes despite the constant rejection in null probes as a function of eccentricity (see above). Nonetheless, t-tests with corrections ($\alpha = 0.02$) confirmed Type 1 AUC remained significantly above chance ($p < 0.001$ for all eccentricities).

### 2.2.2.3. Metacognitive accuracy on present + original versus null discriminability

To test if the objective task performance was accompanied with significant metacognitive access, we quantified subjective task performance using Type 2 AUCs (figure 3b: blue line). Type 2 AUC also decreased significantly as eccentricity increased (table 1). Yet, for all eccentricities, the Type 2 AUCs were significantly above chance ($M = 0.67 \pm 0.02$, one-sample t-test, all $p < 0.001$, with corrections for multiple comparisons).

### 2.2.3. Original versus modified object patch differentiation

### 2.2.3.1. Overlapping response pattern for original and modified object patches

The second aim for the experiment was to test participants' ability to discriminate subtler distinctions, using the manipulated natural images [17]. In these manipulated images, an object onto which a

Table 1. Results of the likelihood ratio tests on the LME models. Effect sizes ($x_1^2$) of fixed-effect predictors for participants' objective (Type 1 AUC) and metacognitive discriminability (Type 2 AUC). *$p < 0.05$, **$p < 0.01$, ***$p < 0.001$, ****$p < 0.0001$.

| | | present + original versus null | original versus modified | | |
| | | eccentricity | eccentricity | congruence | E × C |
|---|---|---|---|---|---|
| Experiment 1 | Type 1 AUC | 25.23*** | 0.59 | 7.97** | 0.18 |
| | Type 2 AUC | 16.57*** | 1.97 | 2.59 | 0.55 |
| Experiment 2 | Type 1 AUC | 298.01*** | 3.44 | 9.62** | 7.24* |
| | Type 2 AUC | 60.80*** | 1.39 | 7.30** | 1.73 |

person is performing an action is replaced with another object, in a way that the replaced object was incongruent with the scene gist.

We first analysed the effects of the congruence in the initial images by comparing the D × C distribution; the second and the third rows of figure 2(i) show the results from the initial congruent and incongruent image trials. Within each panel, we used blue lines to depict the responses for the original object, while red lines for the modified object.

Visual inspection of the data reveals three points. First, the response patterns differ depending on the congruence of the initial image (i.e. 2nd versus 3rd rows). Second, for the initial incongruent image condition (3rd row), the modified and original patches were endorsed with very similar D × C distributions. Third, for the initial congruent image condition (2nd row), the original and modified patches were reported as 'seen' mostly. Yet, there appears to be a difference in the proportion of highest confidence response (i.e. D × C = 4). Our observations were statistically verified. Two-sample $t$-tests revealed the proportion of D × C = 4 responses were significantly different for congruent initial images ($p < 0.001$ for average as well as eccentricity = 0 and 6.5, figure 2(i)e–g). Such difference was not observed in other D × C to the congruent (figure 2(i)e–h) and all D × C to the incongruent initial images (figure 2(i)i–l) (all $p > 0.001$).

### 2.2.3.2. Objective discriminability between original versus modified patches

Next we analysed the discriminability between the original versus modified probe patches with Type 1 AUC (figure 3c,e: blue lines). For this analysis, we repeated the same procedure replacing present + original and null patches with original and modified patches, respectively. We confirmed the three above-mentioned impressions. The average Type 1 AUC across eccentricities was 0.62 (SEM = 0.02, significantly above chance at $p < 0.001$ with $t$-test) for the congruent, and 0.55 (SEM = 0.02, $p = 0.007$) for the incongruent initial images. The performance was higher for the congruent than incongruent images ($p = 0.005$ with paired-sample $t$-test). Compared to present + original versus null patch differentiation (figure 3a: blue line), original versus modified patch differentiation was much less accurate ($p < 0.001$ for both the congruent and incongruent initial images). Unlike the present + original versus null patch differentiation, the original versus modified patch differentiation was not dependent on eccentricities. As shown in table 1, congruence had a significant main effect on participants' Type 1 AUC but the effects of eccentricity and the interaction term were not significant.

### 2.2.3.3. Metacognitive discriminability in original versus modified patch discrimination

We repeated the analysis with Type 2 AUC on the original versus modified patch discrimination (figure 3d,f: blue lines). For the congruent images, Type 2 AUC across eccentricities was 0.55 (SEM = 0.02, significantly above chance, $p = 0.017$ with $t$-test). It was not significantly above chance for the incongruent images (Type 2 AUC = 0.53, SEM = 0.02, $p = 0.05$ with $t$-test). The Type 2 AUCs between congruent and incongruent images were not significantly different ($p = 0.303$ with $t$-test). LME analysis did not find any significant main effect for eccentricity, congruence or interaction (table 1).

### 2.2.4. Exploratory analysis of the effects of each image and its manipulation on the D × C responses

To summarize the results regarding the image congruence and the patch modification, we found better original versus modified patch discrimination performance for congruent initial images, compared to

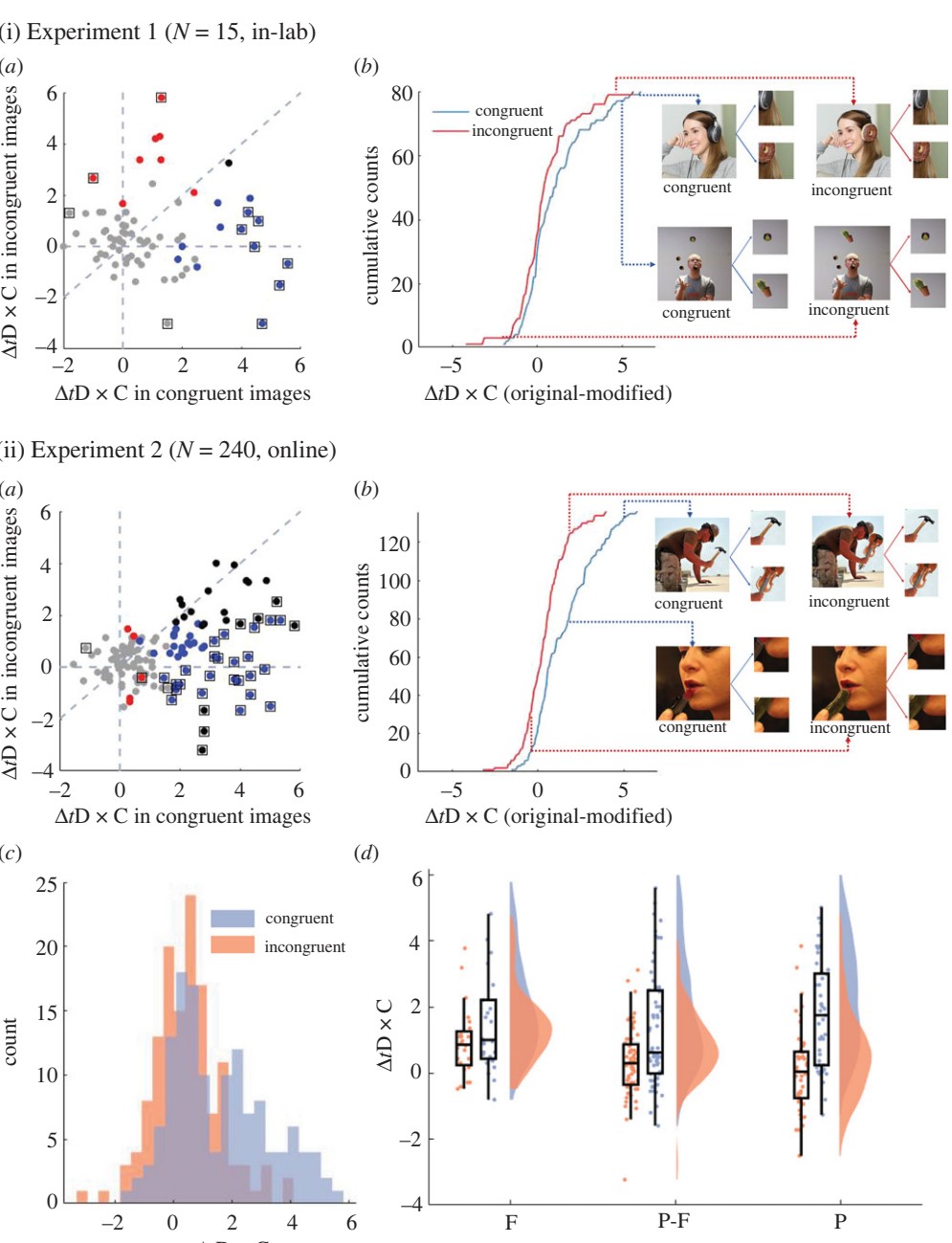

**Figure 4.** Image-based analysis of the effects of congruence of the initial images and object modification in Experiment 1 (i) and Experiment 2 (ii). Based on transformed $D \times C$ ($tD \times C$), we define $\Delta tD \times C$ as mean $tD \times C$ for the original probe − mean $tD \times C$ for the modified probe for each image pair. (i-*a* and ii-*a*) Scatterplots of $\Delta tD \times C$ for the congruent (*x*-axis) and the incongruent (*y*-axis) initial images. Each dot represents the $\Delta tD \times C$ for one image pair ($N = 15$ participants, with $N = 3$ to 12 for each image within the pair for Experiment 1 (i-*a*) and $N = 30$, with $N = 15$ each image for Experiment 2 (ii-*a*)). Four colors indicate the results of the statistical test (two-sample *t*-tests, $p < 0.05$, uncorrected). Grey: no significant difference between original and modified patch $tD \times C$ for both congruent and incongruent initial images. Blue: the difference between original and modified $tD \times C$ only significant for the congruent. Red: only significant for the incongruent. Black: significant for both. We also represented image pairs whose polarity of $\Delta tD \times C$ are opposite (i.e. a significant interaction between patch response type (original/modified) and image congruence in two-way ANOVA) with dots with square outlines. There was no correlation between the congruent and the incongruent initial image in Experiment 1 (i-*a*) ($R^2 = 0.03$, $p = 0.89$) and a weak but significant correlation in Experiment 2 (ii-*a*) ($R^2 = 0.10$, $p < 0.001$). (i-*b* and ii-*b*) Cumulative histogram for $\Delta tD \times C$. The blue and red lines are the $\Delta tD \times C$ for the congruent and incongruent initial images. Here we show two exemplar image pairs with their corresponding $\Delta tD \times C$ (indicated by the source point of the arrows) as well as the original and modified probe patches. (ii-*c*) Histogram distribution of $\Delta tD \times C$ in Experiment 2. (ii-*d*) Fitted distributions, boxplots and scatterplots of $\Delta tD \times C$ against eccentricity levels (F, fovea; P-F, para-fovea; P, periphery) in Experiment 2.

incongruent initial images. In addition, we found different $D \times C$ response patterns on congruent and incongruent initial images. Specifically, there was no difference between the *original* and *modified* patches when the initial images were incongruent (figure 2(i), third row). For the congruent initial images, we found a significant difference between *original* versus *modified* patches in the highest confidence rating ($D \times C = 4$). Before making further conclusions, we performed exploratory image-pair-based analysis. This analysis was guided by our impression during visual inspection of the entire image set. We felt that some image manipulations involved large changes in pixel sizes and close to the fovea (e.g. from headphone to doughnut) which would change the overall semantic of the scene, while others were more subtle, possibly not noticeable with a brief glance of the image. See electronic supplementary material S1 in our Stage-1 report (available at https://osf.io/6w9br/) for all image pairs we used.

To investigate variabilities in $D \times C$ across images, we transform $D \times C$ values by adding 0.5 if $D \times C$ is negative and subtracting 0.5 if positive. The transformed $D \times C$, which we call $tD \times C$ hereafter, ranges from $-3.5$ to $+3.5$ with the uniform interval of 1.

Electronic supplementary material, figure S2, presents all image pairs together with the mean difference of $tD \times C$ values. That is, $\Delta tD \times C(image) = $ (mean across participants $tD \times C$ for the original patch) $-$ (mean across participants $tD \times C$ for the modified patch).

As we presented both the *modified* and the *original* patches to a given participant in Experiment 1, one value of $\Delta tD \times C$ was obtained for either the congruent or the incongruent initial image in a within-subject manner. (Note that in Experiment 2, $\Delta tD \times C(image)$ is defined only across participants as we did not present both the modified and the original patches to a given participant; see below.)

This image-based analysis revealed a rather huge variability in $\Delta tD \times C$, which represents the effect of object manipulation, across image sets. Figure 4(i)a shows the scatter plot of $\Delta tD \times C$ for the congruent and the incongruent initial images (each dot represents the mean $\Delta tD \times C(image)$, averaged across participants). First thing to note is a cluster of dots around the origin. In other words, there were many image pairs that did not differ in the way participants rated $D \times C$ between the *original* or the *modified* image patch (that is, $\Delta tD \times C \sim 0$), regardless of whether the original images were congruent or incongruent. One possible interpretation is that the manipulations in these images are too subtle to notice in a brief glance and do not change image gist (see General discussion).

We quantified this impression by *t*-tests on $tD \times C$ values of each image per congruence condition (with participants as a source of the variance). As the number of available participants per image varied from 3 to 12 for a given congruent or incongruent initial image, these statistics should be taken as exploratory analysis without corrections for multiple comparisons. We show the image pairs whose original and modified patch $tD \times C$ were not significantly different at $p > 0.05$ level (two-sample *t*-tests) for both the congruent and incongruent images, resulting in $\Delta tD \times C \sim 0$ on both axes, with grey dots; these counted 53 out of 80 images (figure 4(i)a). This trend was even clearer from the cumulative histogram in figure 4(i)b. Here, roughly 40 congruent and 50 incongruent images resulted in the $\Delta tD \times C$ value of around 0.

As shown by figure 4(i)a,b, participants' response patterns were highly variable across different image pairs, ranging from a tendency to endorse the modified patch, and reject the original patch (i.e. negative $\Delta tD \times C$), similar responses to both the original and modified patch (i.e. $\Delta tD \times C = 0$), to endorse the original, and reject the modified (i.e. positive $\Delta tD \times C$) and these patterns exist for both initial congruent and incongruent images.

These results tend to imply that we need to carefully evaluate our second question (i.e. the effects of congruence on how much differentiation we can make upon a brief viewing of an image) depending on the image and its manipulation. Our inspection of the image pairs and their $\Delta tD \times C$ (figure 4(i)a,b, as well as electronic supplementary material S1, Stage-1 report at https://osf.io/6w9br/) suggests a factor of size of manipulation and eccentricity of the manipulated object, which has been also noted in previous inattentional blindness literature [4,33]. To perform proper analysis, however, Experiment 1 is not powered enough; thus we propose Experiment 2 below.

## 2.3. Discussion on Experiment 1

In Experiment 1, we tested participants' ability to discriminate what they saw from what they did not; more specifically, they distinguished patches from the natural images that they had briefly viewed from patches that were not part of the viewed images. We used two types of patches for what they did not see. The first type of patches were '*null* patches', which were randomly selected from a different image set [25] and therefore irrelevant to the viewed images. The second type were '*modified*

object patches' which replaced the original object that a person in the image was acting on with another object in a way to make the image incongruent in the scene gist.

Our key results in Experiment 1 were twofold. First, participants demonstrated high objective and metacognitive accuracy in differentiating between *present* and *null* patches (figure 2(i): first row; figure 3*a,b*: blue lines). Second, depending on the congruence of the initial image, we found differences in the distributions of D × C (figure 2(i): second and third rows) as well as the objective discriminability between the *original* versus *modified* patches (figure 3*c,e*: blue lines). Yet, our exploratory analysis (figure 4(i)*a,b*) implies that general conclusion on the effects of image manipulation requires further experiments.

### 2.3.1. Introduction to experiment 2

To solidify our conclusions, we performed Experiment 2 as a registered report. Here, we list the limitations of Experiment 1 and how we hoped to resolve them in Experiment 2.

#### 2.3.1.1. Equalizing number of present + original and null patches

One of the potentially interesting findings apparent from figure 2(i) was that the asymmetry of responses to *present + original* and *null* patches. While participants' responses for *present + original* patches tended to change as a function of patch eccentricity (orange lines in figure 2(i)), their responses for *null* patches tended to be constant across eccentricity (pink lines in figure 2(i)). This is also reflected by the eccentricity-dependent changes in accuracy measures in figure 3*a*. While this has potentially interesting interpretations as to our peripheral perceptions revealed by the MRP, we have to be careful. Specifically, the unequal number of present and null patches might have induced a decision bias to reject patches in general [34]. In Experiment 2, therefore, we presented equal numbers of *null* patches and patches from the presented images, namely six patches per trial, including two present patches, three *null* patches and one *original/modified* object patch.

#### 2.3.1.2. Testing more participants on each image

In Experiment 1, we tested 15 participants, each completed 80 trials, generating 1637 patch responses. Because no study has used the same method before, we could not perform any power analysis to decide the number of participants necessary to examine whether the effects we found in Experiment 1 are generalizable to a wider population. While results presented as figures 2(i) and 3 are statistically significant, our image-based analyses in figure 4(i)*a,b* are severely limited. While the scatter plot (figure 4(i)*a*) suggests a huge heterogeneity of the resulting pattern across images, we had 3–12 participants per image-pair when the initial image was either congruent or incongruent. Based on the results in Experiment 1 and pilot for Experiment 2, we estimate that $N = 30$ participants per image pair ($N = 15$ for congruent and $N = 15$ for incongruent) is enough to detect the significant image specific effects of both the image congruence and the object manipulation. For efficient data collection (and in part due to COVID restriction in face-to-face experiments), we performed Experiment 2 online. In our experience, online experiments work better by shortening the task, unlike two one-hour sessions as in Experiment 1 [35]. Given this, we proposed Experiment 2 to collect the data from 240 participants (i.e. 60 participants per image pair × 136 image pair/34 images per participant).

## 3. Experiment 2

### 3.1. Methods

Experiment 2 was the same as Experiment 1, except for the following aspects.

#### 3.1.1. Participants

In Experiment 2, we tested our patch discrimination task on 240 online participants (18–35 years old). In order to collect data more efficiently, we recruited participants from both Amazon M-Turk and Prolific. To ensure we have valid data, we continuously ran the experiment until we reached 240 participants with catch trial accuracy higher than 0.38 (see Catch trials section for explanation on this cut-off). The study was approved by Monash University Human Research Ethics Committee.

### 3.1.2. Apparatus

The experiment task was programmed on Inquisit 6 [36], and published online via Amazon M-Turk. Participants accessed and completed the task on their own laptops. Before beginning the experiment, participants followed a calibration procedure to make sure the stimuli viewing sizes were consistent with Experiment 1.

### 3.1.3. Stimuli

In Experiment 1, all 15 participants viewed the same 82 pairs of congruent and incongruent images. In Experiment 2, we used 139 pairs of images developed by Shir *et al*. [17]. To calculate the viewing size of images for each participant, we used the data from the calibration. On average, the size of the original image was estimated to be 15.4 (±4.1) dva, slightly smaller than Experiment 1 (19.4 dva). Given the uncertainty in display size estimates, we categorized eccentricity into three levels: fovea (location 5), para-fovea (location 2, 4, 6, 8) and periphery (location 1, 3, 7, 9).

### 3.1.4. Procedure

#### 3.1.4.1. Calibration

Before starting the experiment, participants followed a procedure to calibrate their stimulus display and viewing sizes. Participants first completed the credit card task [37], which can obtain the screen resolution by calculating logical pixel density (i.e. number of pixels per mm).

Then, the participants were asked to hold their arm straight, adjust his/her head position until the width of his/her thumb matched with the length of the line presented at the centre of the screen (electronic supplementary material S2 in Stage-1 report, available at https://osf.io/6w9br/). Here, we used a simple rule that the width of one's thumb held at arm's length is approximately 2 dva [38]. We also asked the participants to complete the blindspot task [37] to measure their viewing distance and the size of their displays. Once the matching was successful, we asked participants to maintain their head position throughout the experiment.

#### 3.1.4.2. Experiment task

After calibration, the experiment began with the same trial procedure as Experiment 1, except for the following. In Experiment 2, for a given image, participants were tested on 6 probe patches, including 2 *present* patches, 3 *null* patches and 1 *original/modified* object patch. We did not present both the *original* and *modified* object patches in one trial. Each participant completed 3 practice trials and 40 experiment trials. The entire procedure took around 30 min including instruction.

We selected 3 image pairs for practice trials. Then we grouped the rest of 136 image pairs into 4 batches (i.e. image pair #1 to #34 for batch 1, #35 to #68 for batch 2, #69 to #102 for batch 3 and #103 to #136 for batch 4). Each participant was randomly assigned to one of these image batches. None of the initial images or probe patches were repeated per participant. Among 34 trials, 17 were congruent and 17 were incongruent trials (in a randomized order). And in each trial, each participant was randomly assigned to view the *original* or *modified* object patch of the initial image.

#### 3.1.4.3. Catch trials

To exclude inattentive participants, we included catch trials. In catch trials (electronic supplementary material S2 in Stage-1 report, available at https://osf.io/6w9br/), the word 'catch', instead of an initial image, was briefly presented (133 ms) on the screen, followed by a response screen, which asked the participant select a particular response option (e.g. 'Yes' with confidence of 4). We included 13 catch trials per participant (one catch trial during the practice). Before data collection, we planned to adopt the accuracy cut-off of 0.38 based on our simulation of random performance (99% quantile).

In summary, throughout an experiment session, the participant completed a calibration procedure, 3 practice trials and 1 practice catch trial, followed by 34 experimental trials and 12 catch trials (randomly intermixed).

### 3.1.5. Data analysis

#### 3.1.5.1. Pilot results

To show the feasibility of testing our paradigm online before Stage 1 acceptance, we ran our task by recruiting 22 participants (14 females) on Amazon M-Turk for a pilot study. For the pilot study, we used image #1 to #40, #41 to #80 and #81 to #120 for $N = 10$, 7 and 5 participants. We show the results in electronic supplementary material S3 of our Stage-1 report (https://osf.io/6w9br/).

#### 3.1.5.2. Data selection

After finishing data collection, we found that due to coding error, there were 10 out of all 8160 trials (240 participants × 34 trials) in which the *original/modified* patches were not presented, but instead randomly replaced with *present* or *null* patches. The 10 trials come from 2 participants (5 trials each) and are for 10 different images. As this coding error only influenced 0.1% of the trials and should not impact the overall results, we decided to include all data into our analysis.

#### 3.1.5.3. Criterion-free analyses on task performance

Following the same procedure described in Experiment 1, we estimated the Type 1 and Type 2 AUC of participants in *present + original* versus *null*, as well as *original* versus *modified* patch discrimination.

#### 3.1.5.4. Linear mixed effects modelling

We performed the same LME modelling procedure on participants' Type 1 and Type 2 AUCs. We only included participants as a random factor because we could not estimate a Type 1/Type 2 AUC for each participant for each image and on each patch eccentricity. This is because AUC is an aggregative measure. We separately addressed the effects of images in our exploratory analysis.

### 3.2. Results

With our online version of the task ($N = 240$, Experiment 2), we replicated the major patterns of the results of the in-laboratory version ($N = 15$, Experiment 1). This includes several central and important questions explained in Introduction. For those aspects that replicated Experiment 1, we do not describe them in detail. Below, we report some differences and extend our analysis in an exploratory manner.

#### 3.2.1. *Present* versus *null* patch discrimination

##### 3.2.1.1. Response patterns of present + original and null patches (first row in figure 2(ii))

We re-examined several observations in Experiment 1, figure 2(i). First of all, participants were highly accurate in endorsing the *present* and *original* patches and rejecting the *null* patches (figure 2(ii)*a–d*). This was despite the fact that we ran the experiment online, where one might expect the performance may not be as good as for in-laboratory participants. Along with previous psychophysics studies [35,39–41], we found that was not the case. We first considered the possibility that Experiment 1 participants showed worse performance because of memory decay (20 probes in Experiment 1 versus 6 probes in Experiment 2, after each image) or fatigue (2 h in Experiment 1 versus 30 min in Experiment 2). However, Experiment 1 participants' responses from the first 6 probes per trial (electronic supplementary material, figure S6I), and the first 24 trials (electronic supplementary material, figure S6II) were very similar to their responses across the entire testing session. Thus, the performance differences between Experiments 1 and 2 are unlikely to be explained by fatigue or memory decay. Several factors may explain better performance in Experiment 2. For example, the equalized numbers of present and absent probes might have led participants to adopt the optimal response criterion. We also improved the instructions for Experiment 2. As opposed to plain text instructions in Experiment 1, Experiment 2 included gif figures for showcasing every step of a trial, which might have improved Experiment 2 participants' understanding of the experiment procedure and resulting in the better performance.

In Experiment 1, we noticed that participants endorse the *present + original* patches less often towards the periphery, while they reject seeing the *null* patches at the same rate across the visual field. We now

interpret this as likely to be due to the different proportions for the *present + original* patches (25% in Experiment 1 versus 50% in Experiment 2) per image. Upon equating them in Experiment 2, these effects were no longer prominent (figure 2(ii)*a–d*). Unlike Experiment 1, participants' proportion of 'no' responses to *null* patches ($x_2^2 = 74.66$, $p < 0.0001$), and 'yes' responses to *present + original* patches (hit) both decreased significantly as eccentricity increases ($x_2^2 = 144.83$, $p < 0.0001$).

### 3.2.1.2. Type 1 and Type 2 AUC analysis

With our criterion-free analysis on objective and metacognitive accuracy, we replicated Experiment 1 (table 1; figure 3*a,bd*: orange lines). In terms of statistical power, our results in Experiment 2 are more reliable due to a larger sample size and more uniform sampling across all images.

With increasing eccentricities, the endorsement of *present + original* patches and the rejection of null patches decreased. This resulted in a decrease in objective accuracy. Using LME modelling, we found a significant main effect of eccentricity in Type 1 AUC (table 1).

To assess the degree of metacognitive access, we calculated the Type 2 AUC. Across patch eccentricities, the average Type 2 AUC was 0.72 (SE = ±0.007, figure 3*b*). Similar to Type 1 AUC, participants' Type 2 AUC also decreased with eccentricity (table 1), although they remain significantly above chance at all eccentricities ($p < 0.0001$).

### 3.2.2. *Original* versus *modified* patch discrimination

#### 3.2.2.1. Response patterns of original and modified patches (in figure 2(ii)e–l)

In Experiment 1 (figure 2(i)*e–h*), upon seeing congruent initial images, participants endorsed seeing the original patch more than the modified patch at the highest confidence rating, regardless of the patch eccentricity. However, upon seeing incongruent images, they did not show any such difference (figure 2(i)*i–l*).

Because we had concerns on the limited data points for each image (3–12 participants per congruent/ incongruent image for each pair), we tried to see if we see the same trends in a larger sample of Experiment 2. By and large, we replicated the results, but with a better statistical sensitivity to uncover further differences.

Across eccentricities, we found that for both congruent and incongruent images, original and modified patches were better differentiated in Experiment 2. For congruent images (figure 2(ii)*e–h*), original and modified patches are not only differentiated on 'yes' decisions with the highest confidence (D × C = 4), but also on 'no' responses on multiple confidence levels (D × C = {−4, −3, −2, 1, and 2}; figure 2(ii)*e–h*). For incongruent images, *original* and *modified* patches were differentiated significantly at fewer confidence levels (D × C = {4, −3 and −4; figure 2(ii)*i–l*).

Surprisingly, for congruent images, participants appeared to discriminate better between *original* and *modified* patches at the periphery than fovea (figure 2(ii)*e–h* and figure 3*c*: orange line). By contrast, for incongruent images, while participants differentiated *original* and *modified* patches significantly on both D × C = 4 and −4, they did so less well as eccentricity increased (figure 2(ii)*i–l*; figure 3*e*: orange line). We follow up these observations with our criterion-free analysis below.

### 3.2.2.2. Objective performance (Type 1 AUC) on original versus modified patches discriminability

Type 1 AUC on original versus modified patch differentiation is plotted in figure 3*c,e* (orange lines). Participants performed better in congruent than incongruent images, both across eccentricity levels and on each eccentricity level. Across eccentricities, participants' mean Type 1 AUC is 0.69 (SEM = 0.01, above chance at $p < 0.0001$ with *t*-test) for the congruent, and 0.56 (SEM = 0.01, $p < 0.0001$) for the incongruent images.

As shown in figure 3*c,e*, eccentricity showed opposite effects for congruent and incongruent images (see the results of LME analysis in table 1). In addition to the significant main effect of congruence, we confirmed the significant interaction between eccentricity and congruence.

To follow up this interaction, we fitted separate LME models on congruent and incongruent Type 1 AUC, each with eccentricity being a fixed-effect predictor, and participants as a random intercept. We did not find a significant effect for eccentricity on both congruent and incongruent Type 1 AUC. Thus, we conclude that participants could differentiate *original* and *modified* patches better for congruent images, while eccentricity had weak effects on discrimination accuracy.

### 3.2.2.3. Metacognitive performance (Type 2 AUC) on original versus modified patches discriminability

In Experiment 1, Type 2 AUC results were not clear, possibly because of a limited and unbalanced number of participants allocated to each congruent/incongruent image. As we expected, the Type 2 AUC results in Experiment 2 were much clearer. Participants obtained a Type 2 AUC of 0.57 (SE = 0.008) for initial congruent images, and 0.53 (SE = 0.008) for initial incongruent images, both above chance at $p < 0.0001$ with $t$-tests. In addition, our LME analysis with likelihood ratio tests revealed a significant main effect of congruence on Type 2 AUC, while the effects of eccentricities and the interaction term were not significant (table 1).

### 3.2.3. Image-based analysis

Experiment 1 implied a huge variance on congruency effects based on each individual image (figure 4(i)a,b). With substantial improvement in statistical power, we investigate the image-based variance in Experiment 2 (figure 4(ii)).

As described for Experiment 1, we first converted D × C into transformed D × C ($t$D × C). Then, we defined

$\Delta t$D × C(image) = (mean across participants $t$D × C for original) − (mean across participants $t$D × C for modified),

separately for initial congruent and incongruent images for each image pair. Because in Experiment 2 each participant was presented with either the *original* or the *modified* patch of an image, $\Delta t$D × C(image) in Experiment 2 is a between-participant variable, rather than within-participant variable in Experiment 1. We plotted the $\Delta t$D × C values in figure 4(ii)a–c.

There were several qualitatively different patterns of the results between Experiments 1 and 2 (figure 4(i),(ii)). Given the reliability of the procedure, we draw conclusions about the image-based analysis mainly from Experiment 2. Even with improved statistical power in Experiment 2, we still observed a huge variance in the effects of congruence across image pairs. As shown in figure 4(ii)a, we found that many image pairs did not show any effect of congruence (81 out of 136 image pairs). This consists of the image pairs in which participants did not differentiate original and modified patches regardless of image congruence (grey dots around 0 in figure 4(ii)a, 66 image pairs), as well as image pairs that participants could differentiate equally well for both congruent and incongruent initial images (black dots along the $x = y$ line, 15 pairs). For image pairs that showed effect of congruence, participants performed better on congruent than incongruent images, with most of the remaining dots falling below the $x = y$ line (figure 4(ii)a). We now turn to potential sources of the large variance in $\Delta t$D × C across images.

### 3.2.3.1. Source of the large variance in $\Delta t$D × C across images

Upon completion of Experiment 1, we speculated that the image-based differences in performance resulted from the differences in critical object sizes and eccentricities. That is, if the original/modified objects are bigger and/or closer to the fovea, they are perceived better, leading to better discrimination performance ($\Delta t$D × C > 0).

As to the eccentricities, plotting $\Delta t$D × C against eccentricity (figure 4(ii)d) suggests that eccentricity might not contribute to the image-based differences. This is formally confirmed by LME analysis in which eccentricity had no significant main effect on $\Delta t$D × C ($x_1^2 = 2.08$, $p = 0.35$).

Next, we analysed the effects of object size. Figure 5a,b explains how we defined the size of the critical object (for details, see electronic supplementary material, figure S3). The raw object size data, which represent the proportion of image the critical object occupies, ranged between 0.003 and 0.39 and were strongly skewed toward 0. To correct for the skewness, we log-transformed the raw object size data before entering them into our subsequent LME analysis. We analysed the influence of object sizes (log-transformed) and congruency on the $\Delta t$D × C variance across images, using the following LME model:

$$\Delta t\text{D} \times \text{C} \sim \text{Size} + \text{Cong} + \text{Size:Cong} + (1 \mid \text{Img}) + (\text{Cong} \mid \text{Img}). \tag{3.1}$$

We estimated a random slope for image congruence on each image pair and a random intercept for each image pair as these effects vary across image pairs (figure 4(ii)).

Figure 5d visualizes the relationship between object sizes and $\Delta t$D × C. As we speculated, critical object sizes were positively correlated with $\Delta t$D × C. We found the significant main effects of size ($x_1^2 = 24.77$, $p < 0.0001$), congruence ($x_1^2 = 19.75$, $p < 0.0001$) and their interaction ($x_1^2 = 5.28$, $p < 0.05$).

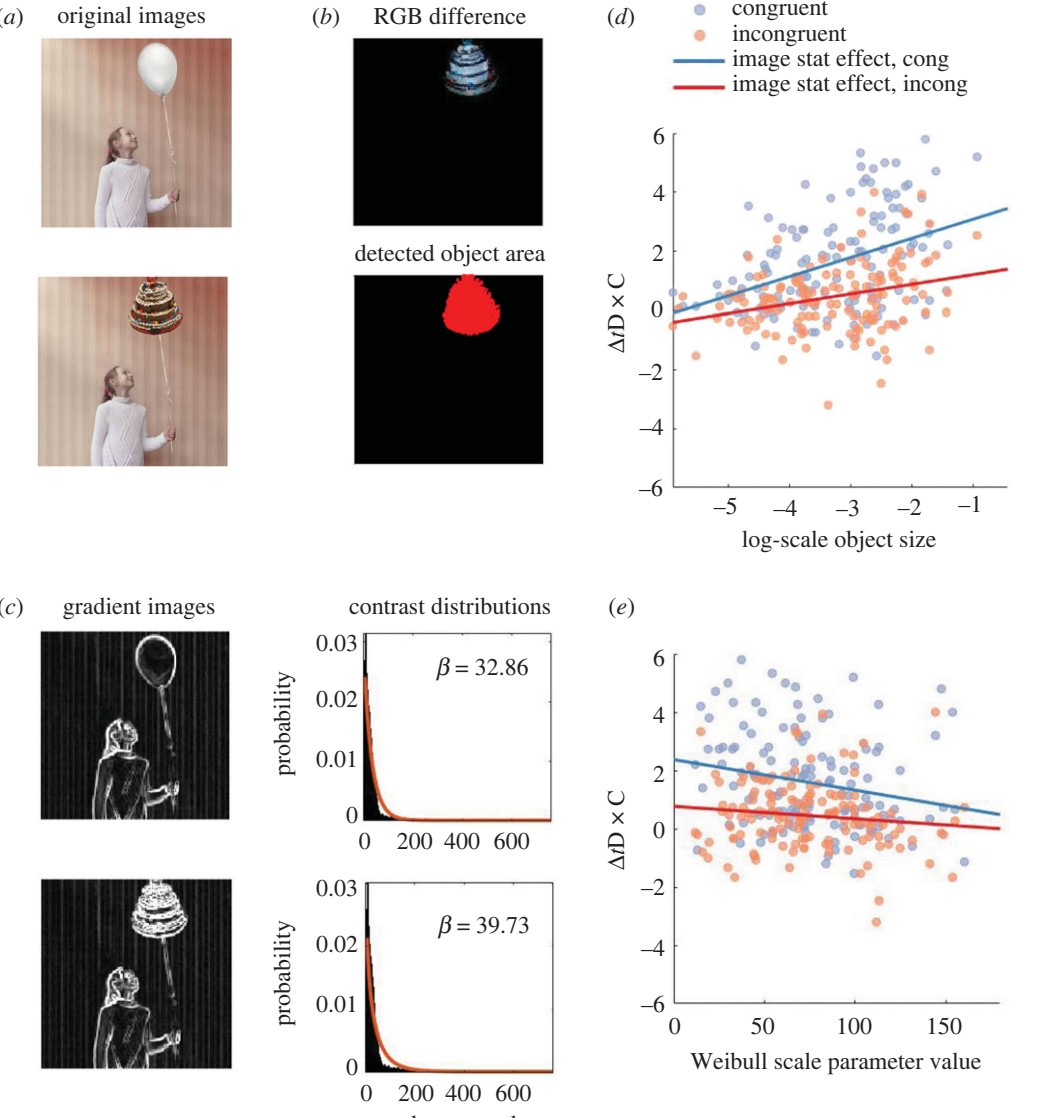

**Figure 5.** Size of the critical object explains the variance across image pairs and the congruence effects on $\Delta tD \times C$. (a) Sample image pair. (b,c) The estimation procedure for object sizes and Weibull scale parameter using the image pair in (a) as an example. To estimate the size of the critical object (b), we obtained the absolute difference in RGB for the two images, from which we detected the area of the object, thresholded as RGB difference = 10. We calculated the proportion of the area in the image as the size of the object. For estimating Weibull scale parameters (c), we filtered the original images to obtain the gradient magnitudes, and then plot the gradient magnitudes (black histograms) as the contrast distributions of the images. We fitted a Weibull distribution for each contrast distribution (orange line), and obtained the scale values $\beta$. For an extended description about the estimation of size and Weibull parameters, see electronic supplementary material, figures S3 and S4. (d) The relation between $\Delta tD \times C$ and object size (log-scaled). Each dot represents an individual image pair. The colour lines represent the best fit regression lines estimated from LME analysis. Blue: congruent initial images. Red: incongruent initial images. (e) The relation between $\Delta tD \times C$ and Weibull scale parameter values, in the same format as (d).

The slope of the best-fit regression was steeper for the congruent (blue) than the incongruent (red) initial images (figure 5d). Separate *post hoc* LME models show significant effects of the size for the congruent images ($x_1^2 = 24.77$, $p < 0.0001$) and the incongruent images ($x_1^2 = 12.01$, $p < 0.001$). Thus, the larger the modified objects, the more easily the differences were noticed, especially when the modified objects were congruent with the initial image gist.

*Post hoc* analysis. Given there are still large variances (figure 4(ii)a), we tested if other image statistics could explain additional variance in $\Delta tD \times C$. We speculated that if the critical objects are embedded in a smoother, as opposed to more cluttered, background, the objects could be recognized and discriminated

more accurately. In addition, we also conjectured that they may be noticed better if either the original or the modified objects 'stood out' with respect to the image background. In such cases, switching between the two objects might induce some awareness, such as 'something was there but now it disappeared' or 'nothing was there but now there is something'.

To test these ideas, we chose Weibull scale parameters [42] to describe the object–background relationship of an image, and difference in visual saliency [43] between the original and modified object patches as an index for the noticeability of object change.

From our LME analysis (electronic supplementary material, table S1), we found that only Weibull scale parameters explained a significant amount of variance in $\Delta t D \times C$, although it had a much weaker effect than size. Therefore, we excluded visual saliency from further analysis. We then followed up on the effect of Weibull parameters, using the same LME model we used for object sizes. We found that both congruence ($x_1^2 = 18.68$, $p < 0.0001$) and Weibull scale parameters ($x_1^2 = 5.99$, $p < 0.05$) had significant main effects on $\Delta t D \times C$. Yet, the interaction between Weibull and congruence was not significant ($x_1^2 = 2.02$, $p = 0.15$). As shown in figure 5e, Weibull scale parameters were negatively associated with $\Delta t D \times C$. This means, as we expected, cluttered images with higher Weibull scale parameters tend to have lower $\Delta t D \times C$. In summary, the variance in $\Delta t D \times C$ across images is largely explained by critical object sizes.

## 3.3. Discussion on Experiment 2

In Experiment 2, we substantially extended Experiment 1 with a larger pool of participants (15 versus 240), which replicated most findings from Experiment 1 with stronger statistical power and generally clearer trends (figures 2 and 3). However, there were a few distinctive patterns of results in Experiment 2, which differed from those observed in Experiment 1.

In Experiment 2, thanks to the increased number of participants per image, our image-based analysis and its follow-up analysis using the size of critical objects and Weibull parameters are highly reliable (figures 4 and 5). We observed that participants showed better discrimination accuracy when the objects are bigger, which is consistent with some inattentional blindness studies [44]. Yet, this is not consistent with previous findings using change detection paradigms, in which the sizes of the changed regions did not impact participants' ability to detect changes in images [45,46]. One explanation of this inconsistency is that in the previous studies, the sizes of the changed regions were relatively small (e.g. 10% of the image at maximum [45]). It is possible that participants could not notice the changed region even when it is of the maximum size. In our study, however, the critical object sizes (i.e. the changed region) took up 0.2% to 39% of the image, so that while the smaller objects remained unnoticeable, the bigger objects might take up a substantial proportion of the images for the participants to notice, resulting in increased discrimination accuracy.

Further, we also observed an interaction between the effect of object sizes and that of object congruence (figure 5d). That is, participants were more accurate in congruent images as the critical objects became bigger. To our knowledge, no study has reported the interaction between object size and congruence. In General discussion, we discuss the implication of this finding in relation to the existing research on scene gist and object recognition.

We also found that the Weibull scale parameter values of images were negatively associated with performance. Weibull scale parameter value is higher when the critical objects are embedded in more cluttered scenes, which would be difficult to notice as we expected. This is also consistent with an idea that object recognition is more difficult when they are embedded in a more cluttered, complex background [47]. While object recognition in a complex scene may require feedback processing [47,48], which typically occurs approximately 150 ms after stimulus presentation [47,49,50], the strong masks we used might have prevented extensive feedback processing.

Unexpectedly, visual saliency difference did not explain much variance across images. This may be because visual saliency was developed for modelling saccadic eye movements in a natural image, which take more than 200 ms [43] (but also see [51,52]). As we presented images for only 133 ms, participants could not make eye movements based on the saliency information of the image. This may explain why visual saliency did not impact their ability to recognize and discriminate the critical objects.

Finally, aggregating participants' responses across images, we observed that participants could differentiate between original and modified objects better, when the critical objects are congruent with the gist of the images. However, when looking at individual congruent/incongruent image pairs, we observed that over half of the image pairs did not show the effect of gist congruence; in other words,

the effect of gist congruence on object discrimination is highly stimulus-dependent. We will discuss the broader implications of this finding in our general discussion.

# 4. General discussion

How informative is a moment of our visual experience in a brief glance? Previous studies attempted to answer this question by probing what human participants can report about the experience they had [53–56]. In this study, we instead explored how well they can discriminate between the actual visual experience they had and many other possible experiences that they could have had. Across Experiments 1 and 2, we found that participants were highly accurate in such differentiation (*present* and *original* versus *null* patches; figure 2(i),(ii), top rows; figure 3a). As we expand below, this high accuracy is not trivial given the set-up of our experiments. With respect to more subtle differences in object changes, the performance was much lower, demonstrating a limitation in such discrimination (figure 3c,e). This subtle object change was not noticed regardless of gist congruence in roughly a half of our image pairs (figure 4(i)a,(ii)a, grey dots). There was also a subset of image pairs in which participants achieved equivalent discrimination accuracy for both the congruent and incongruent images (figure 4(ii)a, black dots along the $x = y$ line). For the rest of the image pairs, we observed the effects of gist congruence, in particular in Experiment 2, where the performance was better when critical objects were congruent with the gist of the image. And this effect interacted with the size of the objects (figure 5d). Finally, we found that along eccentricities, *present* and *original* versus *null* differentiation decreased in both Experiments 1 and 2 (top rows in figure 2(i),(ii); figure 3a). In the following sections, we will discuss the conceptual implications of our paradigm and experiment design, and our key findings.

## 4.1. Conceptual and methodological novelties of the massive report paradigm

In this study, we attempted to test the informativeness of conscious visual experience, by asking participants to differentiate between what they see and what they do not. We summarize the novelties of our paradigm.

### 4.1.1. Conceptual novelty: the definition of information

Many studies measured the information limit of a moment of experience, by asking participants to report what they *saw* from a briefly presented stimuli or quantifying distinguishability between what they saw from its slightly modified version (for examples, see [54–56], and also [53] for a review). This formulation of information captures the degree of visual details that observers capture at a glance. Different from this, we defined the information of conscious experience as the discriminability of how well the actual experience can be differentiated from other possible experiences [57,58]. While the latter idea of information has been proposed for a while, there has been no recent empirical attempts to capture this with psychophysics. Our study is a step towards quantifying the informativeness of experience in this latter sense. The view of informativeness in this sense may be counterintuitive. For example, the experience of looking at a black screen without any perceptual contents is immensely informative, because the observer can easily report a huge number of things that they did not see [58]. Yet, such an idea is compatible with our finding of the high discriminability between *present + original* and *null* patches, which shows participants could easily differentiate between what they saw and what they did not from briefly viewed images.

### 4.1.2. Methodological novelty of massive report paradigm

In the last century, there were some attempts to test the discriminability of experience [59,60] in a manner that is compatible with our concept of informativeness. The most famous study along this line is Sperling's partial report paradigm [61]. Here, we will expound a conceptual parallel and distinctions between Sperling's and our study.

In Sperling' study, participants first viewed an array (e.g. 4 × 3) of alphabet letters very briefly (50 ms, no masking). Then, they heard a ringtone, which cued them to report the letters in one row of the array. He found that participants could report nearly all the letters in any row that was cued, suggesting that participants might have seen and experienced all the letters in the array at a glance before cueing (note

that this interpretation remains controversial and our study does not shed light on this issue; for this issue, see [62–64]).

Our paradigm can be considered as an extension of Sperling's experiment using natural stimuli. But this extension contains several non-trivial components as we elaborate below.

### 4.1.2.1. Natural scene images as stimuli and test probes

The use of natural scene images introduces a huge departure from the original studies in at least three ways.

First, the use of natural images allows us to examine the capacity of human vision at its full capacity in a more ecological way than any other stimuli of a single category (e.g. alphabet letters). Excellent capacity for human vision has been revealed with natural objects in natural scenes in rapid perception [15,65–67], less dependency on attentional amplification [68], and higher visual working memory [28,69]. Most importantly, natural scenes are inherently rich in semantically meaningful contents, which participants can grasp rapidly. Using natural scenes allows us to fathom the truer informativeness in a moment of visual experience.

Additionally, with natural scene images, we can investigate the influence of various perceptual and conceptual factors on the discriminability of experience. Specifically, using the current image set, we can compare participants' ability to differentiate between present and absent contents that are drastically different (present and original versus null patches), as well as the finer-grained differences (original versus modified objects). We can also test other perceptual factors, such as eccentricity [70] and the size of objects [45,46]. As objects are embedded in a semantic context, we can also examine how context congruence interacts with and influences object discrimination [16,18–21,71].

Finally, natural scene stimuli provide us with a virtually infinite number of alternative experiences as null patches. For this study, we used patches randomly cut from a different image set as null patches. Importantly, these null patches have similar statistical properties to present patches. If we were to use any artificial single category of stimuli, like a set of alphabets, the situation is quite different. Alternative experiences are much more limited. These limited alternatives directly lead to an erroneous conclusion that participants can make only a very limited amount of distinction in a brief moment. By taking our novel approach with natural stimuli, participants were able to demonstrate their truer capacity in perceptual discrimination (e.g. not only report that it is not other letters, but that it is also not any other visual experiences, such as seeing red or seeing an animal [13]). Under this condition, we believe we can approach a closer approximation to the immense discriminability of visual experience than Sperling's paradigm, which is a topic of a follow-up theoretical paper we are working on.

### 4.1.2.2. Non-repeating, unexpected stimuli

Another stark departure from Sperling's paradigm is that we did not repeat stimuli to minimize confusability. When alphabets are used repeatedly across trials, participants expect them, which may facilitate or hinder the responses. Repeated stimuli generate expectation and confuse participants [63]. In de Gardelle's study, for example, a strong expectation of the stimuli led participants to report unexpected pseudo-letters as real letters. Using natural scene stimuli, Endress & Potter [72] also showed that expectation from repetitions can hinder memory, called prospective interference. In our study, participants could not expect the content of stimuli as we never repeated the stimuli. Under this condition, our participants demonstrated highly accurate discrimination between what they saw from what they did not. This provides evidence that previous estimation of limited information capacity [59–61] might have resulted from the experimental confounds of stimulus repetition [63,72], combined with the use of artificial stimuli [73]. With ecologically valid stimuli and minimal prospective memory interference, information capacity of a moment of visual experience is much bigger than previously estimated.

### 4.1.2.3. Comparisons/relationships with visual short-term memory tasks

Above, while we discussed the information capacity of a moment of visual experience, our participants were actually asked to view a stimulus, and then report about what they just saw after the stimulus disappeared. This task structure has been used to probe the capacity of visual short-term memory (VSTM) [54,55,74,75]. Which components of the VSTM is our paradigm tapping into when probing the capacity of information in a moment of experience?

The VSTM has been dissected into at least four components: (1) iconic memory [76], (2) fragile VSTM [77,78], (3) large-capacity, short-term memory for semantically meaningful images [72,74,79,80], and (4) visual working memory (VWM) [54,55]. Sperling's task has been regarded to involve iconic memory, an initial high-capacity memory storage coming from the retinal persistence of the stimulus [76]. As our task also required holding the real-time visual input for the cognitive tasks at hand [81], it should reflect the capacity of VWM.

It is relatively easy to rule out the reliance of VWM for our task for two reasons. Firstly, our probe task began immediately after the target image, which left little time for the participants to rehearse and consolidate the initial image into their working memory. Furthermore, the capacity of VWM has been argued to be 4–5 items at maximum [54,55,82]. With the probe patches presented in a serial manner (i.e. 20 and 6 patches for Experiments 1 and 2), we expect each new patch would replace the older items in VWM, severely disrupting the VWM of the initial image by the fourth or fifth probe. However, electronic supplementary material, figure S6, shows that Experiment 1 participants' D × C responses for the first 6 patches versus all patches were highly similar. If our task primarily tapped on the capacity of VWM, we should have observed a marked performance decay for patches presented later in the trial, which we did not observe.

Next, iconic memory and fragile VSTM can be also ruled out by our employment of strong masks. Conventionally (and theoretically), these two memory systems have been considered to be highly fragile and easily erased by visual masking. Strong masks were applied to both initial images and each probe in our task (figure 1). Thus, the high information capacity that we reported in our paradigm cannot be explained by the high capacity of iconic memory or fragile VSTM.

Finally, what about the large-capacity, short-term semantic memory for images [72,74,79]? In these studies, each image is masked by the following images in the sequence, yet large-capacity memory about conceptual understanding of the images has been demonstrated [72,74,79]. Recent study by Thunell & Thorpe [80] examined the capacity limit with a presentation rate up to 120 Hz and a sequence of up to thousands of images.

A critical difference between these studies and ours is the nature of probes. In the previous studies [72,74,79,80], participants' recognition capability was probed with full-sized images presented for at least 500 ms. Thus, each probe can be fully conceptually processed unlike our probe patches. Our small patch probes are not sufficient to recover the global visual and semantic features of the whole image, such as overall structure of the scene and presence of a large object, which are claimed to be crucial for rapid scene recognition [83,84]. Thus, participants in our paradigm likely have relied on perceptual rather than conceptual, and localized rather than global, comparisons between the probe and the viewed initial image.

Overall, the conceptual novelty of our paradigm is threefold. (1) We define the informational capacity of a moment of experience by the discriminability of what participants saw from what they did not. (2) Using natural stimuli, we did not repeat any image and probes, minimizing the effects of expectation and prospective interference. (3) Using cut probe patches, we limited the reliance of global visual and semantic features, uncovering a novel component of the VSTM. This new component has a large capacity (like iconic/fragile VSTM), survives strong masks and remains stable over a long time (like conceptual VSTM or VWM). Yet, its capacity reflects more of the direct and perceptual contents, rather than semantic understanding, of conscious visual experience.

## 4.2. The role of gist in object differentiation task

In this experiment, we examined the influence of the gist of images on critical object differentiation. The interaction between scene gist and the recognition of objects in the scene has been reported since the 1980s [16,18–21,71,85–87]. Broadly speaking, these studies report approximately 10% accuracy advantage when the briefly viewed (approx. 100 ms) object was congruent with the gist of the scene, compared to when it was incongruent.

There are two opposing models that explain this phenomenon. *Functional isolation model* [88–91] proposed that objects and scene context information are processed separately; gist-congruent objects appear to be recognized more accurately because the observers are biased toward attributing a gist-related label to the target object. By contrast, *top-down regulation model* proposed that the processing of the gist of the scene promotes the perceptual analysis of gist-related objects, while suppressing the recognition of unrelated objects; as a result, congruent objects are perceived better than incongruent ones [92,93].

Our results partially support the *top-down regulation* model [92,93] but appear more consistent with the *functional isolation model* [88–90].

Support for the top-down model comes from the better performance for congruent than incongruent initial images (for the congruent initial images: Type 1 AUC = 0.62 ± 0.02 and 0.69 ± 0.01 for Experiments 1 and 2, respectively; for the incongruent initial images: Type 1 AUC = 0.55 ± 0.02 and 0.56 ± 0.01 for Experiments 1 and 2). This suggests that participants could actually see and discriminate critical objects better when they are congruent with the scene gist, which supports the *top-down regulation* model [92,93].

However, support for the *functional isolation model* comes from image-based analysis (figure 4) and interaction between object size and congruency effect (figure 5*d*).

First, when looking at individual image pairs, we observed that less than half of the image pairs showed the effect of semantic congruence (figure 4(i)*a*,(ii)*a*). In Experiment 2 in.particular, 81 out of 136 pairs showed no difference due to congruence (figure 4(ii)*a*, grey dots and the black dots along the *x = y* line). While these images were crafted carefully and appear strikingly different upon long inspection, the critical objects in these images were discriminated equally poorly (grey dots) or well (black dots along the *x = y* line), regardless of the congruence.

Second, figure 5*d* shows that bigger objects were differentiated better in general, and they elicited a stronger accuracy advantage for congruent initial images. This result appears inconsistent with the predictions from the *top-down regulation* model. According to this model, the recognition of ambiguous objects should rely more on the semantic context, predicting the stronger effect of semantic congruence for smaller and more ambiguous objects [94]. But this was not the case.

The latter two observations are more in line with the *functional isolation model* [88–90], where object recognition is independent from the processing of the scene gist, so that an object needs to be perceived as an individual object before it can be processed conceptually [95]. While the size-dependent congruency effect (figure 5*d*) is particularly challenging to the top-down model, we did find some effects of the semantic congruence, unlike the functional isolation model predicts (figure 4). Overall, we conclude that whether congruence directly influences the perception of objects is highly stimulus-dependent; neither *functional isolation* nor *top-down regulation* models could fully explain our results.

## 4.3. The influence of eccentricity on discrimination performance

In our study, we analysed discrimination performance as a function of patch eccentricity. Because we set the size of each patch the same, the proportion of allocated cortical areas should decrease according to the cortical magnification factor at the early visual areas [96,97]. Assuming the discrimination in our task is limited by the responses of the neurons in the early visual areas as in other visual discriminations (e.g. contrast sensitivity [98], object recognition [99,100]; for a review, see [101]), we expected the performance to decline towards the periphery. As expected, as patch eccentricity increased, performance declined in both objective accuracy (e.g. *present* and *original* versus *null*: 10% and 7% for Experiments 1 and 2; figure 3*a*) and metacognitive accuracy (figure 3*b*), which is consistent with an analytical property of high correlation between objective and metacognitive accuracies [102].

From a perspective of conscious phenomenology, the central motivation of our paper, these findings are not trivial. For example, some philosophers invoke the physiological differences between foveal and peripheral vision are often to make claims about a substantial decline in the acuity (or 'detailedness'/'sharpness') in the periphery, which leads to a hypothesis that our feeling of a (rich peripheral) visual experience is merely an illusion [53,103,104]. In our experiment, however, participants' objective accuracy remained much above chance in the periphery (Type 1 AUC = 79% for Experiment 1 and 88% for Experiment 2). This is despite the fact that initial images were completely unexpected, unrepeated natural scenes presented for just 133 ms followed by strong masks, with multiple 133 ms probes asking about the parts of the initial image. Critically, this excellent performance comes with reliable conscious access, measured by our metacognitive measures; participants knew what they saw and what they did not in a conscious reflective manner (figure 3*b*). Given these, we claim it is not easy to consider the illusory nature of visual experience especially in the periphery.

Instead, our findings are generally consistent with an alternative view in that we are experiencing what we should be experiencing across the visual field [13]. According to the recent modelling results by Haun [105], the colourfulness and sharpness of peripheral vision does not decline substantially compared to the fovea, even after considering all the physiological differences between fovea and periphery, such as cortical magnification. Future experiments employing our MRP can further reveal various properties of peripheral discrimination (e.g. summary statistics [53,106]) together with the limit of conscious access.

## 4.4. Future directions

As the first study to propose the MRP, our study explores the influences of various factors, such as eccentricity, object sizes, and semantic congruence, on the discriminability of experience. Here, we discuss some of the outstanding follow-up questions for future experiments.

To begin with, we found that participants' patch discrimination accuracy decreased along eccentricity. While this is consistent with known physiology such as cortical magnification [96,101], the extent of performance decrease (approx. 10%) may be smaller than expected [53,103,104]. This leads to two follow-up questions on cortical magnification. Firstly, if we match the cortical magnification of foveal and peripheral patches by magnifying the sizes of peripheral patches accordingly, will participants achieve equal or better objective accuracy in the periphery? Further, if we were able to equate objective accuracy by matching cortical magnification, what would participants' metacognitive accuracy along eccentricity be? Previous studies [107–109] claim that when objective accuracy is matched, participants demonstrate worse metacognitive accuracy for peripheral stimuli, with a bias to judge peripheral stimuli as seen. And yet those studies used repeated artificial stimuli such as Gabor patches. It is unclear whether the same results can be replicated with natural stimuli that human vision is more adapted to [15,28,65–69], and without stimulus repetition, which minimizes the confounds of expectation [63] and memory interference [72].

We can further explore the mechanism of visual discrimination in the periphery using variations of the current paradigm. For example, to test whether participants used summary statistics of peripheral patches [53,106], we can include some scrambled patches that preserve only the summary statistics of the present patches in the current paradigm (for example, using Portilla & Simoncelli's method [110]). If participants cannot differentiate between the scrambled and the original present patches in the periphery, then it is likely that participants base their judgements on peripheral patches on summary statistics.

## 5. Conclusion

Our study introduced a novel paradigm to test the informativeness of visual experience at a glance. Rather than probing only what is seen, we explored the relations of one experience to other possible experiences—that is, what is seen and what is not seen. As an initial attempt to describe conscious experience in such a novel way, we had a range of interesting findings: participants are highly accurate in discriminating between what they see and what they do not. Discrimination between individual objects was primarily dependent on the object sizes; however, dependent on individual images, semantic congruence between objects and scenes can also influence discrimination performance. The MRP can quantitatively reveal the truer limit of immense informativeness of a moment of consciousness, which is likely to be much richer than previously thought.

Ethics. We have obtained ethical approvals for both Experiment 1 (laboratory-based, psychophysics experiments; project code 10994) and Experiment 2 (online psychophysics experiments; project code 17674) from Monash University Human Research Ethics Committee.

Data accessibility. We have registered our project on Open Science Framework and uploaded our data and codes to Dryad. OSF: https://osf.io/6w9br/ [111]. Dryad Digital Repository: https://doi.org/10.5061/dryad.931zcrjjt [112].

Additional material is provided in the electronic supplementary material [113].

Authors' contributions. L.Q.: conceptualization, data curation, formal analysis, investigation, methodology, project administration, visualization, writing—original draft, writing—review and editing; R.M.G.: supervision, writing—review and editing; N.T.: conceptualization, funding acquisition, methodology, resources, software, supervision, writing—review and editing.

All authors gave final approval for publication and agreed to be held accountable for the work performed therein.

Conflict of interest declaration. We declare we have no competing interests.

Funding. L.Q., R.M.G. and N.T. were supported by the Australian Research Council (DP180104128 and DP180100396). N.T. was also supported by the National Health and Medical Research Council (APP1183280) and Grant-in-Aid for Transformative Research Areas (B) (HDM5Y) from Japan Society for the Promotion of Science.

Acknowledgements. We would like to thank Liad Mudrik for her comments and suggestions on the manuscript, and Ruitong Fan for his assistance in building up the online experiment protocol for this study.

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
