## [Peer Review File · Royal Society Open Science]

Review History

RSOS-210394.R0 (Original submission)

Review form: Reviewer 1 (Adrien Doerig)

Do you have any ethical concerns with this paper?

No

Recommendation?

Accept in principle

Comments to the Author(s)

Liang et al. propose a new paradigm, called Massive Report Paradigm (MRP) to address shortcomings in previous studies on the richness of conscious perception. Specifically, they argue that previous studies only allowed simplistic reports of experience, such as "red vs. green", which does not exhaust what we perceive when we see the stimulus (we may perceive different

shades of red, the shapes of the stimuli, etc, none of which are reported in classic studies). In the MRP proposed to remedy this shortcoming, naturalistic images are presented to a (large number of) subjects, followed by a mask. Then patches of natural images are presented, and subjects have to report if the patch was part of the original image or not. Then the accuracy of observers to report “what they saw” vs. “what they did not” is quantified. In addition, the authors propose that if observers were able to recognize objects incongruent with the scene in this paradigm, that would indicate rich conscious perception (because they can perceive more than just the gist of a scene). There has already been an experiment with interesting results but also limitations identified by the authors. They suggest to conduct a large scale experiment online to face these challenges. Although I expect the discussion of results to be somewhat controversial (but this is a matter for later), I think the MRP is an interesting and relevant paradigm, and the authors seem thorough in their analyses. Therefore, I suggest accepting this proposal. Below are a few suggestions for improving clarity and a more theoretical issue.

Major comments:

- L.216: Is there any reason why all participants got the same order of images? Usually, order is randomized to control for effects of the sequence of stimuli (e.g. maybe subjects are worse for early images, etc.).
- Range corrected DxC: Why is the original [-4.4] scale corrected to [-3.5,3.5]. It sounds strange because an initial value of -0.2 will be mapped to +0.3, while an initial value of 0.2 will be mapped to -0.2. In other words, this procedure inverts some scores. Is there any good reason for this? Also I don't get what the authors mean by “with the uniform interval of 1”. Lastly, the DxC values reported in the results extend to 4, not 3.5. So I must be missing something here, which deserves to be clarified.
- The authors link this experiment to the rich vs. sparse consciousness debates. Can we see more than we can report? Specifically, they propose that if observers were able to recognize objects incongruent with the scene in this paradigm, that would indicate rich conscious perception (because they can perceive more than just the gist of a scene). However, it is important to note that, if observers are able to report incongruent objects, that does not mean they can perceive more than they report (since they report it). Hence, this would not disprove theories of consciousness arguing for “sparse” consciousness where we do not perceive more than we can report. This does not bear on the experiments detailed by the authors, but will be important in what I expect the discussion part to be about.

Minors :

- I am not sure how to understand the first sentence of the abstract. Does it mean “can we differentiate what we see from what we cannot see”, i.e., can we know which things we see and which we do not? Or does it mean “can we differentiate between things we see, and between things we cannot see”, i.e., given that we see something (or not), can we perceive the difference between that thing and other things? In other words, is the paper about which things are consciously perceived or not, or about differences between perceived things. I presume this will become clear later, but it is best if the first sentence is clear already.
- L.42: typo: there is a “ missing in “what it is like”
- On line 27, the authors say that neither probes nor patches were repeated for participants. But on lines 140-141 they say that they tested 15 participants, and 20 questions per image. How can images not be repeated, if there are more questions for each image than participants? Line 27 is a little confusing in this respect.
- L.330: I don't understand this sentence: “the original and modified patches were reported as “seen” mostly with difference in the proportion of highest confidence response (i.e., $D \times C = 4$).” Does this mean that there was only a significant difference between seen and unseen when using the $D \times C = 4$ analysis? And with other DxC there is no difference?

Review form: Reviewer 2

Do you have any ethical concerns with this paper?

No

Recommendation?

Major revision

Comments to the Author(s)

p3, lines 91-99

in general: it took me considerable effort to understand the actual paradigm based on the text until this point. Given that the novelty of the paradigm is key to this paper, I would recommend to give some attention to a clearer explanation of the paradigm here in the introduction.

"if experience is rich ... possible experiences"

Yes, but this does not necessarily mean that *conscious* experience is required!

p3., 1103-104

Scene gist is allegedly also reflected in low level image statistics (see e.g. Groen et al., (2013), <https://www.jneurosci.org/content/33/48/18814.short>); I will get back later to this.

p5. 1186 - 1196. The probes are presented sequentially, and the entire sequence can take up to 45s. How does the paradigm account for image decay over 45s? It would suppose that an image/experience has been stored in working memory in order to perform this task, but it is this transfer of an image to working memory that is believed by some to require attention (and thus not reflect actual conscious experience)...

p7. 1275. It would be really helpful if the authors could just specify the model in Wilkinson notation in stead of a textual description. Additional question: why is probe order not included in the model? This might give important information re: temporal profile of decay of the original image.

p8-10 (results). I would really appreciate the model evaluations to be presented in tables; this makes it a lot easier to parse what is going on.

p10. 1383. The fact that large differences are found between the stimuli does indeed suggest there are relevant differences between the images. Could these differences have something to do with low level image statistics? Has this been checked? And have the authors run the LME with image as random factor (if not -> should definitely be done, see Judd, Westfall & Kenny, 2012; <http://jakewestfall.org/publications/JWK.pdf>). If indeed image statistics do play a role, this would give the authors the possibility to add e.g. Weibull parameters as a fixed factor in the analysis (see Scholte et al., 2009: <https://ivi.fnwi.uva.nl/isis/publications/2009/ScholteJV2009/ScholteJV2009.pdf>).

p12 1467. Experiment 2 is going to be run online; this is going to be an additional challenge with regard to timing. I do not know Inquisit well enough, but I would strongly recommend that the authors include some kind of measure to check presentation times of the target stimuli, and to capture total trial length, in particular if re-analysis of the data of experiment 1 shows a decay effect .

Decision letter (RSOS-210394.R0)

Dear Ms Liang

On behalf of the Editors, I am pleased to inform you that your Manuscript RSOS-210394 entitled "How much can we differentiate at a brief glance: Revealing the truer limit in conscious contents through the Massive Report Paradigm (MRP)" deemed suitable for in-principle acceptance in Royal Society Open Science subject to minor revision in accordance with the referee and editor suggestions. Please find their comments at the end of this email.

The reviewers and handling editors have recommended publication, but also suggest some minor revisions to your manuscript. Therefore, I invite you to respond to the comments and revise your manuscript.

Please you submit the revised version of your manuscript within 7 days (i.e. by the 09-Jun-2021). If you do not think you will be able to meet this date please let me know immediately.

Full author guidelines can be found here <https://royalsocietypublishing.org/rsos/registered-reports#ReviewerGuideRegRep>.

Kind regards
Professor Chris Chambers
Royal Society Open Science
openscience@royalsociety.org

Associate Editor Comments to Author (Professor Chris Chambers):
Comments to the Author:

Thank you for your patience during this challenging time for reviewers. Two expert reviewers have now assessed the manuscript. As you will see their evaluations are largely positive, while also noting a range of areas that would benefit from improvement, including clarity and justification of various aspects of the experimental procedures and analysis plans. Broadly, however, the proposal is strong and well supported by pilot data, therefore provided the authors are able to respond very thoroughly to all points raised, I will assess a

revision at desk and in-principle acceptance should be forthcoming without requiring further in-depth Stage 1 review.

Reviewer comments to Author:

Reviewer: 1

Comments to the Author(s)

(NOTE: I also attach this review in .docx format, which will be easier to read)

Liang et al. propose a new paradigm, called Massive Report Paradigm (MRP) to address shortcomings in previous studies on the richness of conscious perception. Specifically, they argue that previous studies only allowed simplistic reports of experience, such as "red vs. green", which does not exhaust what we perceive when we see the stimulus (we may perceive different shades of red, the shapes of the stimuli, etc, none of which are reported in classic studies). In the MRP proposed to remedy this shortcoming, naturalistic images are presented to a (large number of) subjects, followed by a mask. Then patches of natural images are presented, and subjects have to report if the patch was part of the original image or not. Then the accuracy of observers to report "what they saw" vs. "what they did not" is quantified. In addition, the authors propose that if observers were able to recognize objects incongruent with the scene in this paradigm, that would indicate rich conscious perception (because they can perceive more than just the gist of a scene). There has already been an experiment with interesting results but also limitations identified by the authors. They suggest to conduct a large scale experiment online to face these challenges. Although I expect the discussion of results to be somewhat controversial (but this is a matter for later), I think the MRP is an interesting and relevant paradigm, and the authors seem thorough in their analyses. Therefore, I suggest accepting this proposal. Below are a few suggestions for improving clarity and a more theoretical issue.

Major comments:

- L.216: Is there any reason why all participants got the same order of images? Usually, order is randomized to control for effects of the sequence of stimuli (e.g. maybe subjects are worse for early images, etc.).
- Range corrected DxC: Why is the original [-4.4] scale corrected to [-3.5,3.5]. It sounds strange because an initial value of -0.2 will be mapped to +0.3, while an initial value of 0.2 will be mapped to -0.2. In other words, this procedure inverts some scores. Is there any good reason for this? Also I don't get what the authors mean by "with the uniform interval of 1". Lastly, the DxC values reported in the results extend to 4, not 3.5. So I must be missing something here, which deserves to be clarified.
- The authors link this experiment to the rich vs. sparse consciousness debates. Can we see more than we can report? Specifically, they propose that if observers were able to recognize objects incongruent with the scene in this paradigm, that would indicate rich conscious perception (because they can perceive more than just the gist of a scene). However, it is important to note that, if observers are able to report incongruent objects, that does not mean they can perceive more than they report (since they report it). Hence, this would not disprove theories of consciousness arguing for "sparse" consciousness where we do not perceive more than we can report. This does not bear on the experiments detailed by the authors, but will be important in what I expect the discussion part to be about.

Minors :

- I am not sure how to understand the first sentence of the abstract. Does it mean "can we differentiate what we see from what we cannot see", i.e., can we know which things we see and which we do not? Or does it mean "can we differentiate between things we see, and between things we cannot see", i.e., given that we see something (or not), can we perceive the difference between that thing and other things? In other words, is the paper about which things are

consciously perceived or not, or about differences between perceived things. I presume this will become clear later, but it is best if the first sentence is clear already.

- L.42: typo: there is a " missing in "what it is like"

- On line 27, the authors say that neither probes nor patches were repeated for participants. But on lines 140-141 they say that they tested 15 participants, and 20 questions per image. How can images not be repeated, if there are more questions for each image than participants? Line 27 is a little confusing in this respect.

- L.330: I don't understand this sentence: "the original and modified patches were reported as "seen" mostly with difference in the proportion of highest confidence response (i.e., $D \times C = 4$)."

Does this mean that there was only a significant difference between seen and unseen when using the $D \times C = 4$ analysis? And with other $D \times C$ there is no difference?

Reviewer: 2

Comments to the Author(s)

p3, lines 91-99

in general: it took me considerable effort to understand the actual paradigm based on the text until this point. Given that the novelty of the paradigm is key to this paper, I would recommend to give some attention to a clearer explanation of the paradigm here in the introduction.

"if experience is rich ... possible experiences"

Yes, but this does not necessarily mean that *conscious* experience is required!

p3., l103-104

Scene gist is allegedly also reflected in low level image statistics (see e.g. Groen et al., (2013), <https://www.jneurosci.org/content/33/48/18814.short>); I will get back later to this.

p5. l186 - l196. The probes are presented sequentially, and the entire sequence can take up to 45s. How does the paradigm account for image decay over 45s? It would suppose that an image/experience has been stored in working memory in order to perform this task, but it is this transfer of an image to working memory that is believed by some to require attention (and thus not reflect actual conscious experience)...

p7. l275. It would be really helpful if the authors could just specify the model in Wilkinson notation in stead of a textual description. Additional question: why is probe order not included in the model? This might give important information re: temporal profile of decay of the original image.

p8-10 (results). I would really appreciate the model evaluations to be presented in tables; this makes it a lot easier to parse what is going on.

p10. l383. The fact that large differences are found between the stimuli does indeed suggest there are relevant differences between the images. Could these differences have something to do with low level image statistics? Has this been checked? And have the authors run the LME with image as random factor (if not -> should definitely be done, see Judd, Westfall & Kenny, 2012; <http://jakewestfall.org/publications/JWK.pdf>). If indeed image statistics do play a role, this would give the authors the possibility to add e.g. Weibull parameters as a fixed factor in the analysis (see Scholte et al., 2009: <https://ivi.fnwi.uva.nl/isis/publications/2009/ScholteJV2009/ScholteJV2009.pdf>).

p12 l467. Experiment 2 is going to be run online; this is going to be an additional challenge with regard to timing. I do not know Inquisit well enough, but I would strongly recommend that the authors include some kind of measure to check presentation times of the target stimuli, and to

capture total trial length, in particular if re-analysis of the data of experiment 1 shows a decay effect .

Author's Response to Decision Letter for (RSOS-210394.R0)

See Appendix A.

Decision letter (RSOS-210394.R1)

Dear Ms Qianchen

On behalf of the Editor, I am pleased to inform you that your Stage 1 Registered Report RSOS-210394.R1 entitled "How much can we differentiate at a brief glance: Revealing the truer limit in conscious contents through the Massive Report Paradigm (MRP)" has been accepted in principle for publication in Royal Society Open Science.

You may now progress to Stage 2 and complete the study as approved. Before commencing data collection we ask that you:

- 1) Update the journal office as to the anticipated completion date of your study.
- 2) Register your approved protocol on the Open Science Framework (<https://osf.io/>) or other recognised repository, either publicly or privately under embargo until submission of the Stage 2 manuscript. Please note that a time-stamped, independent registration of the protocol is mandatory under journal policy, and manuscripts that do not conform to this requirement cannot be considered at Stage 2. The protocol should be registered unchanged from its current approved state, with the time-stamp preceding implementation of the approved study design. We strongly recommend using the dedicated RR registration portal at <https://osf.io/rr>

Following completion of your study, we invite you to resubmit your paper for peer review as a Stage 2 Registered Report. Please note that your manuscript can still be rejected for publication at Stage 2 if the Editors consider any of the following conditions to be met:

- The results were unable to test the authors' proposed hypotheses by failing to meet the approved outcome-neutral criteria.
- The authors altered the Introduction, rationale, or hypotheses, as approved in the Stage 1 submission.
- The authors failed to adhere closely to the registered experimental procedures. Please note that any deviations from the approved experimental procedures must be communicated to the editor immediately for approval, and prior to the completion of data collection. Failure to do so can result in revocation of in-principle acceptance and rejection at Stage 2 (see complete guidelines for further information).
- Any post-hoc (unregistered) analyses were either unjustified, insufficiently caveated, or overly dominant in shaping the authors' conclusions.

- The authors' conclusions were not justified given the data obtained.

We encourage you to read the complete guidelines for authors concerning Stage 2 submissions at <https://royalsocietypublishing.org/rsos/registered-reports#ReviewerGuideRegRep>. Please especially note the requirements for data sharing, reporting the URL of the independently registered protocol, and that withdrawing your manuscript will result in publication of a Withdrawn Registration.

Once again, thank you for submitting your manuscript to Royal Society Open Science and we look forward to receiving your Stage 2 submission. If you have any questions at all, please do not hesitate to get in touch. We look forward to hearing from you shortly with the anticipated submission date for your stage two manuscript.

on behalf of Professor Chris Chambers (Registered Reports Editor, Royal Society Open Science)
openscience@royalsociety.org

Author's Response to Decision Letter for (RSOS-210394.R1)

See Appendix B.

Decision letter (RSOS-210394.R2)

Dear Ms Qianchen:

It is a pleasure to accept your manuscript entitled "How much can we differentiate at a brief glance: Revealing the truer limit in conscious contents through the Massive Report Paradigm (MRP)" in its current form for publication in Royal Society Open Science.

Thank you for your fine contribution. On behalf of the Editors of Royal Society Open Science, we look forward to your continued contributions to the journal.

on behalf of Professor Chris Chambers (Subject Editor)
openscience@royalsociety.org

Appendix A

Associate Editor Comments to Author (Professor Chris Chambers):

Comments to the Author:

Thank you for your patience during this challenging time for reviewers.

Two expert reviewers have now assessed the manuscript. As you will see their evaluations are largely positive, while also noting a range of areas that would benefit from improvement, including clarity and justification of various aspects of the experimental procedures and analysis plans.

Broadly, however, the proposal is strong and well supported by pilot data, therefore **provided the authors are able to respond very thoroughly to all points raised, I will assess a revision at desk and in-principle acceptance should be forthcoming without requiring further in-depth Stage 1 review.**

Dear Prof Chris Chambers,

We are delighted to receive this largely positive Stage 1 evaluation from two reviewers. Below, we include the point-by-point replies for all points raised by the reviewers. By incorporating various suggestions, we improved the clarity throughout and justified various plans. We hope that our manuscript is now ready for in-principle acceptance and move on to the next stage of data collection.

Regards

Liang Qianchen and Nao Tsuchiya

Thank you for your comments. Below, we copied your original comments in Italic and provided our reply in normal fonts.

Reviewer comments to Author:

Reviewer: 1

Comments to the Author(s)

(NOTE: I also attach this review in .docx format, which will be easier to read)

Liang et al. propose a new paradigm, called Massive Report Paradigm (MRP) to address shortcomings in previous studies on the richness of conscious perception. Specifically, they argue that previous studies only allowed simplistic reports of experience, such as “red vs. green”, which does not exhaust what we perceive when we see the stimulus (we may perceive different shades of red, the shapes of the stimuli, etc, none of which are reported in classic studies). In the MRP proposed to remedy this shortcoming, naturalistic images are presented to a (large number of) subjects, followed by a mask. Then patches of natural images are presented, and subjects have to report if the patch was part of the original image or not. Then the accuracy of observers to report “what they saw” vs. “what they did not” is quantified. In addition, the authors propose that if observers were able to recognize objects incongruent with the scene in this paradigm, that would indicate rich conscious perception (because they can perceive more than just the gist of a scene). There has already been an experiment with interesting results but also limitations identified by the authors. They suggest to conduct a large scale experiment online to face these challenges. Although I expect the discussion of results to be somewhat controversial (but this is a matter for later), I think the MRP is an interesting and relevant paradigm, and the authors seem thorough in their analyses. Therefore, I suggest accepting this proposal. Below are a few suggestions for improving clarity and a more theoretical issue.

Thank you for your fair evaluation!

Major comments:

- L.216: Is there any reason why all participants got the same order of images? Usually, order is randomized to control for effects of the sequence of stimuli (e.g. maybe subjects are worse for early images, etc.).

Authors' reply: We admit that this feature of Experiment 1, all images presented in the same order for all participants, was an unintended feature of the designed task. Experiment 1 was the first behavioural task that the first author (LQ) designed independently, and her supervisors, RG and NT, failed to notice the fixed order across participants until the data analysis. Having said that, we do not see this feature as invalidating the result of Experiment 1. In addition, we plan to conduct Experiment 2 by randomizing the order of the images for each participant, as stated in our registered report manuscript. This allows us to compare and contrast its results with Experiment 1, to check if there are any effects of the fixed sequence of the images.

- Range corrected Dx_C: Why is the original [-4.4] scale corrected to [-3.5,3.5]. It sounds strange because an initial value of -0.2 will be mapped to +0.3, while an initial value of 0.2 will be mapped to -0.2. In other words, this procedure inverts some scores. Is there any good reason for this? Also I don't get what the authors mean by “with the uniform interval of 1”. Lastly, the Dx_C values reported in the results extend to 4, not 3.5. So I must be missing something here, which deserves to be clarified.

Authors' reply: Dx_C means the multiplication of D (yes/no decision) and C (confidence). As D can take -1 or 1 and C can take 1, 2, 3, and 4, Dx_C values can only take one of 8 discrete integers (-4, -3, -2, -1, 1, 2, 3, 4) without 0. And yet the distance between Dx_C = 1 and Dx_C = -1 is 2, while all adjacent integers have a distance of 1. Therefore, we mapped the original [-4, -1] to [-3.5,-0.5] and [1, 4] to [0.5,3.5]. As a result, rcDx_C values take only one of 8 numbers, (-3.5, -2.5, -1.5, -0.5, 0.5,

1.5, 2.5, 3.5). We now realize the term “range correction” is misleading as it indicates uniform shrinking of the values towards 0. Thus, we replace the term with tDxC (transformed DxC) and modify the relevant part of the manuscript, as follows:

P.6, L. 211 - 214: To represent a response for one probe patch, we encoded “yes” decision as “+” and “no” as “-” and multiply it with the confidence level (1, 2, 3 and 4), to obtain decision x confidence (DxC) value, which can take one of 8 integers (-4, -3, -2, -1, 1, 2, 3, 4).

P. 10, L. 380 - 384: ...because the distance between $DxC = 1$ and $DxC = -1$ is 2, while that between other adjacent alternatives is 1, we transformed DxC values (tDxC) by adding 0.5 if DxC is negative and subtracting 0.5 if positive. . The tDxC therefore ranged from -3.5 to +3.5 with the uniform interval of 1.

- *The authors link this experiment to the rich vs. sparse consciousness debates. Can we see more than we can report? Specifically, **they propose that if observers were able to recognize objects incongruent with the scene in this paradigm, that would indicate rich conscious perception (because they can perceive more than just the gist of a scene)**. However, it is important to note that, **if observers are able to report incongruent objects, that does not mean they can perceive more than they report (since they report it)**. Hence, this would not disprove theories of consciousness arguing for “sparse” consciousness where we do not perceive more than we can report. This does not bear on the experiments detailed by the authors, but will be important in what I expect the discussion part to be about.*

Authors' reply: We would like to clarify that in our manuscript, we did not write “if observers were able to recognize objects incongruent with the scene in this paradigm, that would indicate rich conscious perception”. What we claimed in L.90-92 as well as L.120-122 is “if conscious perception is rich, the observers should be able to recognize incongruent objects”. Logically speaking, therefore, what the reviewer says is the converse of our claim. In fact, we do agree with the reviewer that “if observers are able to report incongruent objects, that does not mean they can perceive more than they report”.

We anticipate a more nuanced discussion upon the Stage 2 manuscript stage.

Minors :

- *I am not sure how to understand the first sentence of the abstract. Does it mean “can we differentiate what we see from what we cannot see”, i.e., can we know which things we see and which we do not? Or does it mean “can we differentiate between things we see, and between things we cannot see”, i.e., given that we see something (or not), can we perceive the difference between that thing and other things? In other words, is the paper about which things are consciously perceived or not, or about differences between perceived things. I presume this will become clear later, but it is best if the first sentence is clear already.*

Authors' reply: Thank you for pointing out this ambiguity. What we meant was the first interpretation. To improve the clarity of the abstract, we have modified the first sentence of the abstract as follows:

P.1, L. 16: “Upon a brief glance, how well can we differentiate what we see from what we do not?”

- L.42: typo: there is a “missing in “what it is like”

Authors' reply: Thanks, we have added the “ in P.1, L.40.

- On line 27, the authors say that neither **probes** nor **patches** were repeated for participants. But on lines 140-141 they say that they tested 15 participants, and 20 questions per image. How can **images** not be repeated, if there are more questions for each image than participants? Line 27 is a little confusing in this respect.

Authors' reply: We apologize for this confusing aspect of our task design and terminologies.

Here we clarify that in each trial, participants are briefly presented with a natural scene **image** only once at the beginning of the trial. Following that, we present each of 20/21 **probe patches** in a sequence without showing the initial image again. If the reviewer believes that participants should not be able to remember the aspect of the image while answering 20/21 probe patch questions, indeed, this is one of the surprising aspects of our experiment.

This figure above shows the percentage of present judgment for each of three probe types (present & original, where we expect present judgment; modified; and null, where we expect absent judgement). Clearly, participants can view the initial image briefly, and continue to respond to the multiple probe patches.

Another clarification. For each participant, we never presented the same **image**. Each of 80 trials used a unique initial image.

Further, across 80 trials x 20/21 **probe patches**, again, we never repeated the same patch twice. We were able to do this because we had access to a large database.

In sum, he/she will only see a specific image, or a specific probe patch once throughout the entire experiment. To avoid confusion from the readers, we have modified the relevant sentence in the abstract to be consistent with the rest of the paper, as follows:

P1, L. 27: "Neither the images nor probe patches were repeated per participant."

To make clear the distinction between initial images and probe patches in our task, we also corrected some of our wordings in the Methods (see highlighted parts in P.5, as well as Figure captions of Figure 1 on P. 19), for example the following:

P.19, L.646 - 648: The procedure to generate **probe patches**. Each of the 9 patches were tagged with a location number 1-9 corresponding to their locations in the **original image**.

- L.330: *I don't understand this sentence: "the original and modified patches were reported as "seen" mostly with difference in the proportion of highest confidence response (i.e., $D \times C=4$)." Does this mean that there was only a significant difference between seen and unseen when using the $D \times C=4$ analysis? And with other $D \times C$ there is no difference?*

Authors' reply: Yes. We have clarified this in P.9, L. 335 - 340, as follows:

The third observation was statistically verified: two-sample t-tests revealed the proportion of $tD \times C=3.5$ responses were significantly different for congruent initial images ($p < .001$ for average and each eccentricity, Fig 4e-h). Such difference was not observed in other $D \times C$ responses to the congruent (Fig 4e-h) and all $D \times C$ responses to the incongruent initial images (Fig 4i-l). (all $p > .001$).

Thank you for your comments. Below, we copied your original comments in Italic and provided our reply in normal fonts.

Reviewer: 2

Comments to the Author(s)

p3, lines 91-99

in general: it took me considerable effort to understand the actual paradigm based on the text until this point. Given that the novelty of the paradigm is key to this paper, I would recommend to give some attention to a clearer explanation of the paradigm here in the introduction.

Authors' reply: We apologize for the ambiguity in our introduction to our experiment paradigm. To better clarify the novelty of our paradigm, we moved what was originally in L.68-81 to after L.73.

"if experience is rich ... possible experiences"

*Yes, but this does not necessarily mean that *conscious* experience is required!*

Authors' reply: We apologize for the ambiguity. In the previous manuscript, when we used the term "experience", we always meant "conscious" experience. To avoid ambiguity, we replaced all "experience" with either "conscious experience" or "conscious perception" in the revision.

Having clarified our terminology, we do not fully understand what the reviewer meant. To quote our original sentence with this clarification what we meant was:

"If conscious experience is rich, then the experiencer should be able to differentiate one conscious experience from a very large number of other possible conscious experiences".

One possible reading of what the reviewer meant would be that even if the stimulus was presented not consciously, participants should be able to differentiate that nonconscious experience from a very large number of other possible nonconscious experiences. While the reviewer may be correct, as it can be the case for blindsight patients, we do not know any literature that is directly relevant for our experiment presented here.

p3., l103-104

Scene gist is allegedly also reflected in low level image statistics (see e.g. Groen et al., (2013), <https://www.jneurosci.org/content/33/48/18814.short>); I will get back later to this.

Authors' reply: Thanks. We replied to the later comment.

*p5. l186 - l196. The probes are presented sequentially, and the entire sequence can take up to 45s. **How does the paradigm account for image decay over 45s?** It would suppose that an image/experience has been stored in working memory in order to perform this task, but it is this transfer of an image to working memory that is believed by some to require attention (and thus not reflect actual conscious experience)...*

Authors' reply: Figure below shows the percentage of "present" judgments participants gave to each type of patches (Present and original patches, modified patches, and null patches), as a function of question number within a trial. As is clear from the figure, participants showed only minor decay in their performance from the first probe patch (x-axis) to the 20th probe patch

throughout a trial. This suggests that the performance of our MRP is unlikely to depend on working memory, which should decay quickly and be limited in capacity up to 4-7 items.

p7. l275. It would be really helpful if the authors could just specify the model in **Wilkinson notation** in stead of a textual description. Additional question: **why is probe order not included in the model?** This might give important information re: temporal profile of decay of the original image.

Authors' reply: Models are now expressed in Wilkinson notation. We revised this aspect as follows:

P.7, L.258 - 270, "To investigate the effect of eccentricity (numerical; x_1) on participants' AUCs for *present + original* vs. *null* patch differentiation (y_1), we built the following LME model on Type 1 and Type 2 AUCs respectively:

$$y_1 \sim x_1 + (x_1 | n) + (1 | n) \quad (1)$$

Here, we included participants (n) a random slope and intercept {Barr, 2013 #201}. In order to investigate the role of eccentricity (x_2) and congruence (x_3) on participants' AUCs for original vs. modified patch differentiation (y_2), we estimated the following LME model on participants' Type 1 and Type 2 AUCs separately:

$$y_2 \sim x_2 + x_3 + x_2 x_3 + (x_2 | n) + (x_3 | n) + (1 | n) \quad (2)$$

We performed LME models on Type 1 and Type 2 AUCs, which are indices of probe discriminability aggregating across 20/21 probe responses per image per participant. Besides, as per our response to the last comment, we did not find substantial performance decay along probe order.

p8-10 (results). I would really appreciate the model evaluations to be presented in tables; this makes it a lot easier to parse what is going on.

Authors' reply: LME model results are now presented in tables (P. 8, Table 1). We have also modified the relevant text on P. 9 - 10 (see highlighted sentences).

p10. I383. The fact that large differences are found between the stimuli does indeed suggest there are relevant differences between the images. **Could these differences have something to do with low level image statistics? Has this been checked? And have the authors run the LME with image as random factor (if not -> should definitely be done, see Judd, Westfall & Kenny, 2012; <http://jakewestfall.org/publications/JWK.pdf>). If indeed image statistics do play a role, this would give the authors the possibility to add e.g. Weibull parameters as a fixed factor in the analysis (see Scholte et al., 2009: <https://ivi.fnwi.uva.nl/isis/publications/2009/ScholteJV2009/ScholteJV2009.pdf>).**

Authors' reply: In fact, we analysed the role of low-level image properties before the original submission, but we concluded that we do not have enough statistical power to be conclusive about any of the analyses we performed. Knowing that we do not have enough statistical power, we have tried adding images as a random slope and intercept in our LME models, but it did not seem to change the effect sizes of the fixed-effect predictors. However, we suspect that with enough statistical power in Experiment 2, we can appropriately test the image based effect, which we will report.

In terms of the other analysis we performed on the data in Experiment 1, we analysed whether participants were more likely to endorse a given null patch, when its low-level image properties (e.g., mean luminance, luminance contrast or contrast in each of three color channels) are similar to those of the present patch. In our preliminary analysis, we did not find clear evidence that such local low-level image properties explain a large proportion of variance in our results.

On the other hand, as we showed in Figure 4, there exists a clear image-based effect. As the reviewer can inspect in Supplementary Material, we anticipate that image-based effects are unlikely to be driven by these low-level image statistics.

Having said this, we realize that this is an important question and we will report the statistics appropriately with enough statistical power upon completion of Experiment 2.

p12 I467. Experiment 2 is going to be run online; this is going to be an additional challenge with regard to timing. I do not know Inquisit well enough, but I would strongly recommend that the authors include some kind of measure to check presentation times of the target stimuli, and to capture total trial length, in particular if re-analysis of the data of experiment 1 shows a decay effect.

Authors' reply: While an online experiment that runs on a browser would heavily rely on the internet connection, Inquisit asks participants to download all components that are necessary to run the task beforehand. Further, Inquisit blocks other applications to be run in parallel. Thus, Inquisit is highly reliable in control of the timing.

In addition, our lab has run several studies similar to Experiment 2, where we compared the performance of 10 participants in the lab and >600 participants online. There we found very good consistency between the participants.

Finally, our pilot data also showed that participants' performance started saturating around 66ms, and stabilized around 100ms. Given that the stimuli duration in the current experiment is 133ms, missing one or two frames (100-166ms) should not affect the results. To ensure that our

Experiment 2 results are not confounded by online testing itself, we will again check the consistency of results between lab- and online-testing, on an image-by-image basis.

Appendix B

Associate Editor Comments to Author (Professor Chris Chambers):

Comments to the Author:

Thank you for your patience during this challenging time for reviewers.

Two expert reviewers have now assessed the manuscript. As you will see their evaluations are largely positive, while also noting a range of areas that would benefit from improvement, including clarity and justification of various aspects of the experimental procedures and analysis plans.

Broadly, however, the proposal is strong and well supported by pilot data, therefore **provided the authors are able to respond very thoroughly to all points raised, I will assess a revision at desk and in-principle acceptance should be forthcoming without requiring further in-depth Stage 1 review.**

Dear Prof Chris Chambers,

We are delighted to receive this largely positive Stage 1 evaluation from two reviewers. Below, we include the point-by-point replies for all points raised by the reviewers. By incorporating various suggestions, we improved the clarity throughout and justified various plans. We hope that our manuscript is now ready for in-principle acceptance and move on to the next stage of data collection.

Regards

Liang Qianchen and Nao Tsuchiya

Thank you for your comments. Below, we copied your original comments in Italic and provided our reply in normal fonts.

Reviewer comments to Author:

Reviewer: 1

Comments to the Author(s)

(NOTE: I also attach this review in .docx format, which will be easier to read)

Liang et al. propose a new paradigm, called Massive Report Paradigm (MRP) to address shortcomings in previous studies on the richness of conscious perception. Specifically, they argue that previous studies only allowed simplistic reports of experience, such as “red vs. green”, which does not exhaust what we perceive when we see the stimulus (we may perceive different shades of red, the shapes of the stimuli, etc, none of which are reported in classic studies). In the MRP proposed to remedy this shortcoming, naturalistic images are presented to a (large number of) subjects, followed by a mask. Then patches of natural images are presented, and subjects have to report if the patch was part of the original image or not. Then the accuracy of observers to report “what they saw” vs. “what they did not” is quantified. In addition, the authors propose that if observers were able to recognize objects incongruent with the scene in this paradigm, that would indicate rich conscious perception (because they can perceive more than just the gist of a scene). There has already been an experiment with interesting results but also limitations identified by the authors. They suggest to conduct a large scale experiment online to face these challenges. Although I expect the discussion of results to be somewhat controversial (but this is a matter for later), I think the MRP is an interesting and relevant paradigm, and the authors seem thorough in their analyses. Therefore, I suggest accepting this proposal. Below are a few suggestions for improving clarity and a more theoretical issue.

Thank you for your fair evaluation!

Major comments:

- L.216: Is there any reason why all participants got the same order of images? Usually, order is randomized to control for effects of the sequence of stimuli (e.g. maybe subjects are worse for early images, etc.).

Authors' reply: We admit that this feature of Experiment 1, all images presented in the same order for all participants, was an unintended feature of the designed task. Experiment 1 was the first behavioural task that the first author (LQ) designed independently, and her supervisors, RG and NT, failed to notice the fixed order across participants until the data analysis. Having said that, we do not see this feature as invalidating the result of Experiment 1. In addition, we plan to conduct Experiment 2 by randomizing the order of the images for each participant, as stated in our registered report manuscript. This allows us to compare and contrast its results with Experiment 1, to check if there are any effects of the fixed sequence of the images.

- Range corrected DxC: Why is the original [-4,4] scale corrected to [-3.5,3.5]. It sounds strange because an initial value of -0.2 will be mapped to +0.3, while an initial value of 0.2 will be mapped to -0.2. In other words, this procedure inverts some scores. Is there any good reason for this? Also I don't get what the authors mean by “with the uniform interval of 1”. Lastly, the DxC values reported in the results extend to 4, not 3.5. So I must be missing something here, which deserves to be clarified.

Authors' reply: DxC means the multiplication of D (yes/no decision) and C (confidence). As D can take -1 or 1 and C can take 1, 2, 3, and 4, DxC values can only take one of 8 discrete integers (-4, -3, -2, -1, 1, 2, 3, 4) without 0. And yet the distance between $D \times C = 1$ and $D \times C = -1$ is 2, while all adjacent integers have a distance of 1. Therefore, we mapped the original [-4, -1] to [-3.5,-0.5] and [1, 4] to [0.5,3.5]. As a result, rcDxC values take only one of 8 numbers, (-3.5, -2.5, -1.5, -0.5, 0.5,

1.5, 2.5, 3.5). We now realize the term “range correction” is misleading as it indicates uniform shrinking of the values towards 0. Thus, we replace the term with tDxC (transformed DxC) and modify the relevant part of the manuscript, as follows:

P.6, L. 211 - 214: To represent a response for one probe patch, we encoded “yes” decision as “+” and “no” as “-” and multiply it with the confidence level (1, 2, 3 and 4), to obtain decision x confidence (DxC) value, which can take one of 8 integers (-4, -3, -2, -1, 1, 2, 3, 4).

P. 10, L. 380 - 384: ...because the distance between $D \times C = 1$ and $D \times C = -1$ is 2, while that between other adjacent alternatives is 1, we transformed DxC values (tDxC) by adding 0.5 if DxC is negative and subtracting 0.5 if positive. . The tDxC therefore ranged from -3.5 to +3.5 with the uniform interval of 1.

- *The authors link this experiment to the rich vs. sparse consciousness debates. Can we see more than we can report? Specifically, **they propose that if observers were able to recognize objects incongruent with the scene in this paradigm, that would indicate rich conscious perception (because they can perceive more than just the gist of a scene).** However, it is important to note that, **if observers are able to report incongruent objects, that does not mean they can perceive more than they report (since they report it).** Hence, this would not disprove theories of consciousness arguing for “sparse” consciousness where we do not perceive more than we can report. This does not bear on the experiments detailed by the authors, but will be important in what I expect the discussion part to be about.*

Authors' reply: We would like to clarify that in our manuscript, we did not write “if observers were able to recognize objects incongruent with the scene in this paradigm, that would indicate rich conscious perception”. What we claimed in L.90-92 as well as L.120-122 is “if conscious perception is rich, the observers should be able to recognize incongruent objects”. Logically speaking, therefore, what the reviewer says is the converse of our claim. In fact, we do agree with the reviewer that “if observers are able to report incongruent objects, that does not mean they can perceive more than they report”.

We anticipate a more nuanced discussion upon the Stage 2 manuscript stage.

Minors :

- *I am not sure how to understand the first sentence of the abstract. Does it mean “can we differentiate what we see from what we cannot see”, i.e., can we know which things we see and which we do not? Or does it mean “can we differentiate between things we see, and between things we cannot see”, i.e., given that we see something (or not), can we perceive the difference between that thing and other things? In other words, is the paper about which things are consciously perceived or not, or about differences between perceived things. I presume this will become clear later, but it is best if the first sentence is clear already.*

Authors' reply: Thank you for pointing out this ambiguity. What we meant was the first interpretation. To improve the clarity of the abstract, we have modified the first sentence of the abstract as follows:

P.1, L. 16: “Upon a brief glance, how well can we differentiate what we see from what we do not?”

- L.42: typo: there is a “ missing in “what it is like”

Authors' reply: Thanks, we have added the “ in P.1, L.40.

- On line 27, the authors say that neither **probes** nor **patches** were repeated for participants. But on lines 140-141 they say that they tested 15 participants, and 20 questions per image. How can **images** not be repeated, if there are more questions for each image than participants? Line 27 is a little confusing in this respect.

Authors' reply: We apologize for this confusing aspect of our task design and terminologies.

Here we clarify that in each trial, participants are briefly presented with a natural scene **image** only once at the beginning of the trial. Following that, we present each of 20/21 **probe patches** in a sequence without showing the initial image again. If the reviewer believes that participants should not be able to remember the aspect of the image while answering 20/21 probe patch questions, indeed, this is one of the surprising aspects of our experiment.

This figure above shows the percentage of present judgment for each of three probe types (present & original, where we expect present judgment; modified; and null, where we expect absent judgement). Clearly, participants can view the initial image briefly, and continue to respond to the multiple probe patches.

Another clarification. For each participant, we never presented the same **image**. Each of 80 trials used a unique initial image.

Further, across 80 trials x 20/21 **probe patches**, again, we never repeated the same patch twice. We were able to do this because we had access to a large database.

In sum, he/she will only see a specific image, or a specific probe patch once throughout the entire experiment. To avoid confusion from the readers, we have modified the relevant sentence in the abstract to be consistent with the rest of the paper, as follows:

P1, L. 27: "Neither the images nor probe patches were repeated per participant."

To make clear the distinction between initial images and probe patches in our task, we also corrected some of our wordings in the Methods (see highlighted parts in P.5, as well as Figure captions of Figure 1 on P. 19), for example the following:

P.19, L.646 - 648: The procedure to generate **probe patches**. Each of the 9 patches were tagged with a location number 1-9 corresponding to their locations in the **original image**.

- L.330: *I don't understand this sentence: "the original and modified patches were reported as "seen" mostly with difference in the proportion of highest confidence response (i.e., $D \times C = 4$)." Does this mean that there was only a significant difference between seen and unseen when using the $D \times C = 4$ analysis? And with other $D \times C$ there is no difference?*

Authors' reply: Yes. We have clarified this in P.9, L. 335 - 340, as follows:

The third observation was statistically verified: two-sample t-tests revealed the proportion of $D \times C = 3.5$ responses were significantly different for congruent initial images ($p < .001$ for average and each eccentricity, Fig 4e-h). Such difference was not observed in other $D \times C$ responses to the congruent (Fig 4e-h) and all $D \times C$ responses to the incongruent initial images (Fig 4i-l). (all $p > .001$).

Thank you for your comments. Below, we copied your original comments in Italic and provided our reply in normal fonts.

Reviewer: 2

Comments to the Author(s)

p3, lines 91-99

in general: it took me considerable effort to understand the actual paradigm based on the text until this point. Given that the novelty of the paradigm is key to this paper, I would recommend to give some attention to a clearer explanation of the paradigm here in the introduction.

Authors' reply: We apologize for the ambiguity in our introduction to our experiment paradigm. To better clarify the novelty of our paradigm, we moved what was originally in L.68-81 to after L.73.

"if experience is rich ... possible experiences"

*Yes, but this does not necessarily mean that *conscious* experience is required!*

Authors' reply: We apologize for the ambiguity. In the previous manuscript, when we used the term "experience", we always meant "conscious" experience. To avoid ambiguity, we replaced all "experience" with either "conscious experience" or "conscious perception" in the revision.

Having clarified our terminology, we do not fully understand what the reviewer meant. To quote our original sentence with this clarification what we meant was:

"If conscious experience is rich, then the experiencer should be able to differentiate one conscious experience from a very large number of other possible conscious experiences".

One possible reading of what the reviewer meant would be that even if the stimulus was presented not consciously, participants should be able to differentiate that nonconscious experience from a very large number of other possible nonconscious experiences. While the reviewer may be correct, as it can be the case for blindsight patients, we do not know any literature that is directly relevant for our experiment presented here.

p3., l103-104

Scene gist is allegedly also reflected in low level image statistics (see e.g. Groen et al., (2013), <https://www.jneurosci.org/content/33/48/18814.short>); I will get back later to this.

Authors' reply: Thanks. We replied to the later comment.

*p5. l186 - l196. The probes are presented sequentially, and the entire sequence can take up to 45s. **How does the paradigm account for image decay over 45s?** It would suppose that an image/experience has been stored in working memory in order to perform this task, but it is this transfer of an image to working memory that is believed by some to require attention (and thus not reflect actual conscious experience)...*

Authors' reply: Figure below shows the percentage of "present" judgments participants gave to each type of patches (Present and original patches, modified patches, and null patches), as a function of question number within a trial. As is clear from the figure, participants showed only minor decay in their performance from the first probe patch (x-axis) to the 20th probe patch

throughout a trial. This suggests that the performance of our MRP is unlikely to depend on working memory, which should decay quickly and be limited in capacity up to 4-7 items.

p7. l275. It would be really helpful if the authors could just specify the model in **Wilkinson notation** in stead of a textual description. Additional question: **why is probe order not included in the model?** This might give important information re: temporal profile of decay of the original image.

Authors' reply: Models are now expressed in Wilkinson notation. We revised this aspect as follows:

P.7, L.258 - 270, "To investigate the effect of eccentricity (numerical; x_1) on participants' AUCs for *present + original* vs. *null* patch differentiation (y_1), we built the following LME model on Type 1 and Type 2 AUCs respectively:

$$y_1 \sim x_1 + (x_1 | n) + (1 | n) \quad (1)$$

Here, we included participants (n) a random slope and intercept {Barr, 2013 #201}. In order to investigate the role of eccentricity (x_2) and congruence (x_3) on participants' AUCs for original vs. modified patch differentiation (y_2), we estimated the following LME model on participants' Type 1 and Type 2 AUCs separately:

$$y_2 \sim x_2 + x_3 + x_2 x_3 + (x_2 | n) + (x_3 | n) + (1 | n) \quad (2)$$

We performed LME models on Type 1 and Type 2 AUCs, which are indices of probe discriminability aggregating across 20/21 probe responses per image per participant. Besides, as per our response to the last comment, we did not find substantial performance decay along probe order.

p8-10 (results). I would really appreciate the model evaluations to be presented in tables; this makes it a lot easier to parse what is going on.

Authors' reply: LME model results are now presented in tables (P. 8, Table 1). We have also modified the relevant text on P. 9 - 10 (see highlighted sentences).

p10. I383. The fact that large differences are found between the stimuli does indeed suggest there are relevant differences between the images. **Could these differences have something to do with low level image statistics? Has this been checked? And have the authors run the LME with image as random factor (if not -> should definitely be done, see Judd, Westfall & Kenny, 2012; <http://jakewestfall.org/publications/JWK.pdf>). If indeed image statistics do play a role, this would give the authors the possibility to add e.g. Weibull parameters as a fixed factor in the analysis (see Scholte et al., 2009: <https://ivi.fnwi.uva.nl/isis/publications/2009/ScholteJV2009/ScholteJV2009.pdf>).**

Authors' reply: In fact, we analysed the role of low-level image properties before the original submission, but we concluded that we do not have enough statistical power to be conclusive about any of the analyses we performed. Knowing that we do not have enough statistical power, we have tried adding images as a random slope and intercept in our LME models, but it did not seem to change the effect sizes of the fixed-effect predictors. However, we suspect that with enough statistical power in Experiment 2, we can appropriately test the image based effect, which we will report.

In terms of the other analysis we performed on the data in Experiment 1, we analysed whether participants were more likely to endorse a given null patch, when its low-level image properties (e.g., mean luminance, luminance contrast or contrast in each of three color channels) are similar to those of the present patch. In our preliminary analysis, we did not find clear evidence that such local low-level image properties explain a large proportion of variance in our results.

On the other hand, as we showed in Figure 4, there exists a clear image-based effect. As the reviewer can inspect in Supplementary Material, we anticipate that image-based effects are unlikely to be driven by these low-level image statistics.

Having said this, we realize that this is an important question and we will report the statistics appropriately with enough statistical power upon completion of Experiment 2.

p12 I467. Experiment 2 is going to be run online; this is going to be an additional challenge with regard to timing. I do not know Inquisit well enough, but I would strongly recommend that the authors include some kind of measure to check presentation times of the target stimuli, and to capture total trial length, in particular if re-analysis of the data of experiment 1 shows a decay effect.

Authors' reply: While an online experiment that runs on a browser would heavily rely on the internet connection, Inquisit asks participants to download all components that are necessary to run the task beforehand. Further, Inquisit blocks other applications to be run in parallel. Thus, Inquisit is highly reliable in control of the timing.

In addition, our lab has run several studies similar to Experiment 2, where we compared the performance of 10 participants in the lab and >600 participants online. There we found very good consistency between the participants.

Finally, our pilot data also showed that participants' performance started saturating around 66ms, and stabilized around 100ms. Given that the stimuli duration in the current experiment is 133ms, missing one or two frames (100-166ms) should not affect the results. To ensure that our

Experiment 2 results are not confounded by online testing itself, we will again check the consistency of results between lab- and online-testing, on an image-by-image basis.